# Unc13A and Unc13B contribute to the decoding of distinct sensory information in *Drosophila*

Atefeh Pooryasin [1], Marta Maglione [1,2], Marco Schubert[1], Tanja Matkovic-Rachid[1],
Sayed-mohammad Hasheminasab[3,4], Ulrike Pech[5,6], André Fiala [5], Thorsten Mielke [7] &
Stephan J. Sigrist [1,2 ✉]

The physical distance between presynaptic $Ca^{2+}$ channels and the $Ca^{2+}$ sensors triggering the release of neurotransmitter-containing vesicles regulates short-term plasticity (STP). While STP is highly diversified across synapse types, the computational and behavioral relevance of this diversity remains unclear. In the *Drosophila* brain, at nanoscale level, we can distinguish distinct coupling distances between $Ca^{2+}$ channels and the (m)unc13 family priming factors, Unc13A and Unc13B. Importantly, coupling distance defines release components with distinct STP characteristics. Here, we show that while Unc13A and Unc13B both contribute to synaptic signalling, they play distinct roles in neural decoding of olfactory information at excitatory projection neuron (ePN) output synapses. Unc13A clusters closer to $Ca^{2+}$ channels than Unc13B, specifically promoting fast phasic signal transfer. Reduction of Unc13A in ePNs attenuates responses to both aversive and appetitive stimuli, while reduction of Unc13B provokes a general shift towards appetitive values. Collectively, we provide direct genetic evidence that release components of distinct nanoscopic coupling distances differentially control STP to play distinct roles in neural decoding of sensory information.

[1] Institute for Biology/Genetics, Freie Universität Berlin, Berlin, Germany. [2] NeuroCure Cluster of Excellence, Charité Universitätsmedizin, Berlin, Germany. [3] Department of Dermatology, Venereology and Allergology, Charité Universitätsmedizin, Berlin, Germany. [4] CCU Translational Radiation Oncology, DKTK, National Center for Tumor Diseases (NCT), Heidelberg University Hospital (UKHD) and German Cancer Research Center (DKFZ), Heidelberg, Germany. [5] Department of Molecular Neurobiology of Behavior, Johann-Friedrich-Blumenbach-Institute for Zoology and Anthropology, University of Göttingen, Göttingen, Germany. [6] Laboratory of Neuronal Communication, VIB Center for the Biology of Disease, K.U.Leuven, Leuven, Belgium. [7] Max Planck Institute for Molecular Genetics, Berlin, Microscopy and Cryo-Electron Microscopy Group, Berlin, Germany. ✉email: stephan.sigrist@fu-berlin.de

In response to external sensory stimuli, sensory neurons change their activities by firing sequences of action potentials (APs) in various temporal patterns. These patterns are then transmitted over their output synapses. Synaptic transmission in turn relies on the rapid fusion of neurotransmitter-containing synaptic vesicles (SVs), which happens in response to AP-induced $Ca^{2+}$ influx at active zones (AZs). A highly conserved molecular machinery cooperates at SV-release sites to mediate SV plasma membrane attachment and maturation, $Ca^{2+}$ sensing, and membrane fusion. Despite this high degree of conservation, synapses—even within the same organism, organ, or neuron—can be highly diverse regarding the probability of APs to trigger SV fusion. Importantly, repetitive activation can lead to either strengthening or weakening of transmission, resulting in a rather tonic or phasic filtering characteristic. This "short-term plasticity" (STP), referring to use-dependent changes in synapse strength on time scales of tens to hundreds of milliseconds, has been demonstrated to be highly diversified across the brain synapses of experimental mammals[1–3].

The molecular basis of STP is the subject of intense investigation. Importantly, evolutionarily conserved AZ-scaffolding proteins can determine the coupling distance between SV fusion sites and voltage-gated $Ca^{2+}$ channels (VGCC) and, thereby, shape STP[4]. Indeed, effective coupling distances vary across mammalian brain synapses, resulting in major functional differences[5,6]. Here, biophysical and electrophysiological analyses suggest that both release probability and STP depend greatly on the nanometer scale distance between SVs with their $Ca^{2+}$ sensor synaptotagmin, and VGCCs[3,5–9]. This is explained by the sharp spatiotemporal profile of AP-induced $Ca^{2+}$ transients and by the fact that SV fusion is activated by the cooperative binding of several (3–5) $Ca^{2+}$ ions, resulting in a strong distance relationship for release probability: SVs positioned closer to the $Ca^{2+}$ source have much higher release probabilities than distant ones. However, upon stimulation with higher AP frequencies, synapses with high release probability tend to depress as the resupply of SVs becomes limiting, resulting in a fast "phasic" release profile. In contrast, synapses with bigger coupling distances display a slower but sustained "tonic" release profile[5,10].

Essential release factors of the m(Unc13) protein family thereby seem to define SV-release sites and position SVs relative to the VGCCs[11]. Understanding and demonstrating the behavioral relevance of coupling diversity and STP is less advanced. We recently showed that in the *Drosophila* brain two Unc13 species, Unc13A and Unc13B, co-exist with Unc13A operating close (≈70 nm) but Unc13B further away (>100 nm) from the VGCCs[12]. By combining biochemical/genetic analysis with STED super-resolution microscopy, we previously showed that physical interactions with specific AZ scaffold proteins mediates these distinct coupling modes, with ELKS family Bruchpilot (BRP) targeting Unc13A but Syd-1 Unc13B[12].

Consistently, at the first relay synapse of the *Drosophila* olfactory system uniquely amenable to electrophysiological analysis across *Drosophila* brain synapses, Unc13A promotes a high probability but depressing phasic, while Unc13B a slower tonic release component[12].

We here set out to test these release components with different STP characteristics in the synaptic decoding of behaviorally relevant information. Due to their "bottleneck" position we targeted excitatory projection neurons (ePNs), the second relay neurons of the olfactory neuronal information stream, and their AZs for this analysis. Both Unc13A and Unc13B contributed to SV-release from ePNs. However, *unc13A* and *unc13B* knockdown had distinct behavioral consequences. While loss of Unc13A equally affected the smell response to either appetitive or aversive odors, loss of Unc13B shifted odor responses to a more appetitive

behavior, regardless of the innate valence of the odor. We further found that the Unc13B-mediated transmission component in inhibitory projection neurons (iPNs) operated antagonistically to Unc13B in ePNs, making the convergence region of both ePNs and iPNs, called lateral horn (LH), the likely place of signal integration here. Taken together, distinct coupling distances of synaptic release components not only are associated with distinct STP behavior and the effective transmission of either tonic or phasic information, but seemingly differently filter sensory information in behaviorally relevant decoding.

## Results

**Unc13A and Unc13B co-exist within AZs of projection neurons.** Individual axons of excitatory projection neurons (ePNs) first project to the mushroom body (MB) calyx, synapsing onto MB neurons, and then to the lateral horn (LH). In contrast, inhibitory PNs (iPNs) bypass the calyx to project directly to the LH (Fig. 1a). In order to specifically and discretely label presynaptic AZs of excitatory and inhibitory projection neurons in the MB calyx and the LH, we expressed *brp-short* $^{GFP}$ [13,14] using the driver lines GH146-Gal4[15] and MZ699-Gal4[16], respectively. BRP-short incorporates into the AZ scaffold of existing synapses using homotypic BRP–BRP interactions, but owing to the dependence of its transport on endogenous BRP hardly forms ectopic signals[12]. We first analyzed Unc13A and Unc13B distribution by immunofluorescence staining with "Airyscan microscopy", a technique that allowed us to sample large volumes at improved resolution[17]. Overall, Unc13A and Unc13B were expressed rather equally over PN terminals in both calyx and LH. The BRP-short $^{GFP}$ label allowed for the analysis on the level of individual presynaptic AZs. Clearly, individual GFP-positive AZs harbored both Unc13A and Unc13B at the PN terminals (Fig. 1b) in both calyx and LH. To validate this impression, we calculated Mander's coefficient values, which were very high between the BRP-short $^{GFP}$ label and both Unc13A and Unc13B (Supplementary Fig. 1). Next, we stained for Drep2[18] to identify the postsynaptic densities of cholinergic synapses. Indeed, individual AZs of ePNs expressed both Unc13A and Unc13B, and consistent with the cholinergic nature of the ePNs, adjacent postsynaptic densities were Drep2-positive (Fig. 1c). In the MB calyx, PN terminals form presynaptic boutons, which are tightly surrounded by the postsynaptic Kenyon cells dendrites ("claws"), collectively referred to as microglomerular organization[19]. We observed a similar pre-post-synaptic organization in the LH as well (Fig. 1c). The expression of the *brp-short* $^{GFP}$ construct specifically in iPNs (using MZ699-Gal4) showed a rather sparse label in the LH, with again individual AZs being co-positive for Unc13A and Unc13B. The Mander's coefficient values, though still high, here showed a somewhat lower level of co-occurrence between BRP-short $^{GFP}$ and Unc13B than BRP-short $^{GFP}$ and Unc13A (Supplementary Fig. 1b). As expected for inhibitory neurons, Drep2 was absent opposite to these AZs (Fig. 1d).

**Unc13A and Unc13B display distinct physical coupling distances in the LH.** At several *Drosophila* brain synapses analyzed so far, Unc13A and Unc13B displayed distinct nano-topologies on the level of individual AZs, relative to their center where $Ca^{2+}$ channels cluster[12,20,21]. Here, Unc13A localized at ~70 nm from the BRP ring center, while Unc13B was found at larger coupling distances (>100 nm), consistent with their differential role in neurotransmission (phasic release behavior for Unc13A versus tonic behavior for Unc13B)[11,12,20,21]. To check whether this nano-domain spacing indeed applied to all types of PN axonal terminals, also at PN AZs within the LH, we analyzed "nano-topologies" within individual AZs with two- or three-channel 2D time-gated STED (gSTED,

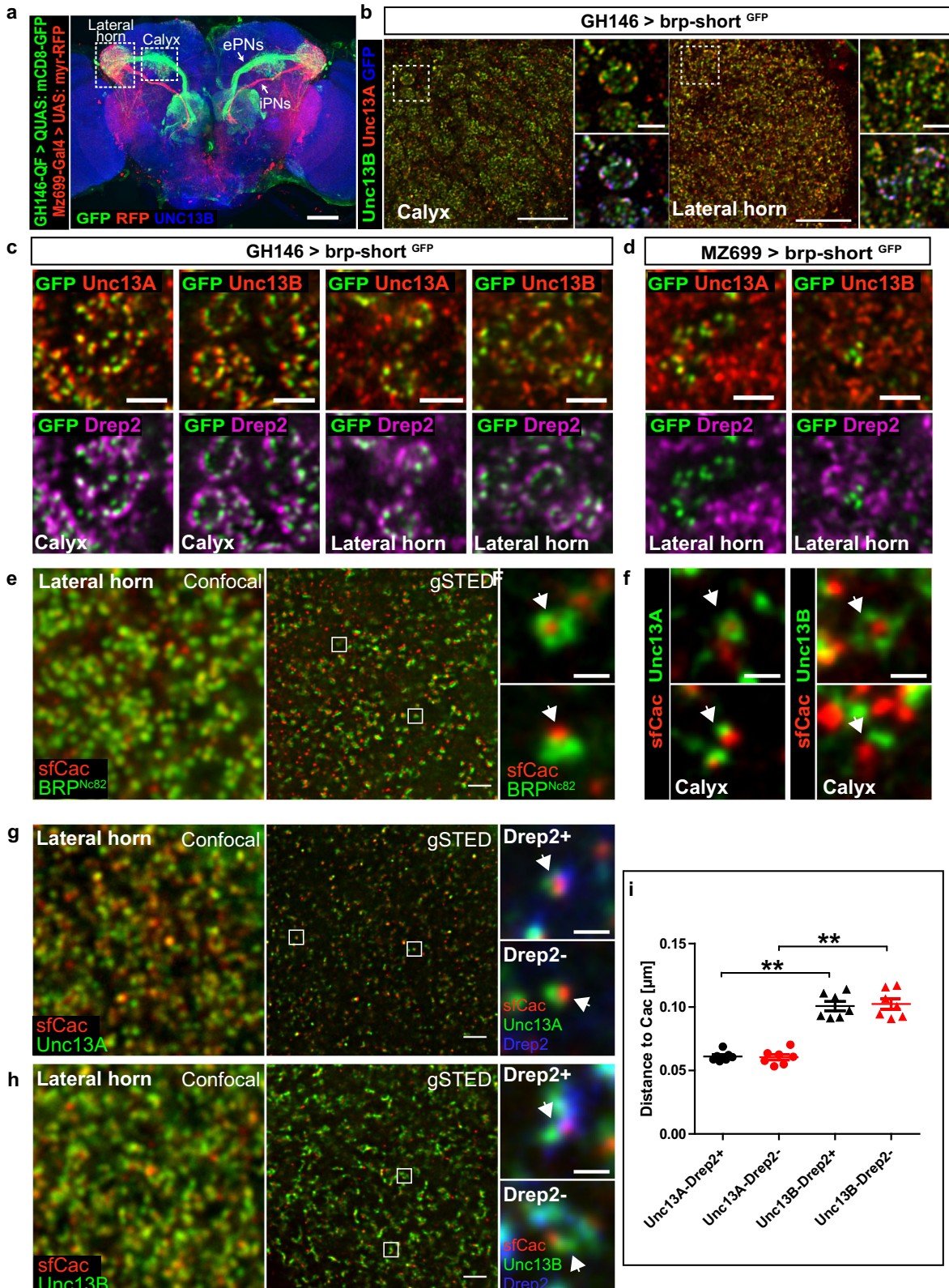

lateral resolution ~40 nm) imaging, using a fly line in which voltage-operated $Ca^{2+}$ channel α1-subunit *Cacophony* was functionally on-locus tagged with GFP (sfCac^{GFP})[22]. gSTED images of planarly imaged AZs in the LH showed $Ca^{2+}$ channel clusters surrounded by rings of scaffold protein Bruchpilot (BRP; NC82 monoclonal anti-body) (Fig. 1e), highly reminiscent of the nano-organization pre-viously observed at PN derived AZs in the calyx or at NMJ AZs[12,20].

We then analyzed Unc13A and Unc13B localization relative to sfCac spots in ePN boutons in the calyx (Fig. 1f) and the LH (Fig. 1g, h and Supplementary Fig. 2). As expected from our earlier study[12], in the calyx, Unc13A localized closer to sfCac^{GFP} clusters than Unc13B (Fig. 1f). In the LH, AZs of ePNs (via GH146 expression) could easily be distinguished from inhibitory neuron-derived AZs, as only ePN AZs, being cholinergic, showed

**Fig. 1 Unc13A and Un13B localization at active zones of ePN and iPN presynapses. a** Expression of myr-RFP in iPNs using MZ699-Gal4 (UAS/Gal4 system) and mCD8-GFP in ePNs using GH146-QF (QUAS/QF system). Image shows maximal projection fluorescence intensity across a confocal stack of a whole brain, immunostained against GFP (green), RFP (red), and Unc13B (blue). Lateral horn, MB calyx, iPN, and axonal tracks are indicated ($n = 2$, experiment was performed once). **b** Expression of UAS: brp-short $^{GFP}$ under control of GH146-Gal4 ($n = 5$, experiment was performed two times). Images represent single focal planes from calyx and LH regions immunostained against Unc13A (red), Unc13B (green), and GFP (blue). **c, d** Expression of UAS: brp-short $^{GFP}$ using GH146-Gal4 (**c**) and MZ699-Gal4 (**d**). Images represent single focal planes from calyx and LH regions immunostained against Unc13A or Unc13B (red), Drep2 (magenta), and GFP (green) ($n = 5$, experiment was performed two times). **e–i** Time-gated STED (gSTED) analysis of Unc13A and B nano-distribution at LH and MB calyx synapses. **e** Confocal (left) and gSTED (right) images of BRP $^{Nc82}$ (green) and the GFP-labeled Ca$^{2+}$ channel subunit sfCac (red) at LH AZs. Insets show a planar AZ with sfCac localizing at the center of BRP rings (upper) and a vertical AZ (lower) ($n = 3$, experiment was performed once). **f** gSTED images of Unc13A or Unc13B (green) and sfCac (red) at PN:KC synapses in the MB calyx ($n = 3$, experiment was performed once). **g, h** Confocal (left) and gSTED (right) images of Unc13A (**g**) or Unc13B (**h**) (green) and sfCac (red) in the LH. Insets show a Drep2-positive excitatory synapse (upper) and a Drep2-negative inhibitory synapse (lower) from triple-channel gSTED images. **i** Mean Unc13A- and Unc13B-sfCac distances at Drep2-positive and Drep2-negative LH synapses ($n = 7$ brains/group). Drep2-positive synapses: Unc13A-sfCac ($61 \pm 1$ nm) versus Unc13B-sfCac ($101 \pm 4$ nm). Drep2-negative synapses: Unc13A-sfCac ($60 \pm 2$ nm) versus Unc13B-sfCac ($102 \pm 4$ nm). Values represent mean ± SEM. Statistics: **$p < 0.01$, Kruskal–Wallis test followed by two-tailed Dunn´s multiple comparison test. White squares: magnified regions. White arrowheads: region of interest. Scale bars: **a** 50 µm, **b** 10 µm, **c–d** 2 µm, **e, g, h** 1 µm, insets: 200 nm, **f** 200 nm. Source data including the exact sample sizes and the $p$ values are provided as a Source Data file. See also Supplementary Fig. 1 and 2.

apposed postsynaptic Drep2 clusters[18] (Fig. 1c, d). We then quantified Unc13A and Unc13B coupling distances relative to Ca$^{2+}$ channels at both Drep2$^+$ and Drep2$^-$ synapses in the LH (Fig. 1g, h). At both synapse types, Unc13A spots were found at ~60 nm from sfCac spots (Fig. 1g–i and Supplementary Fig. 2), whereas Unc13B spots localized at ~100 nm relative to the sfCac signals (Fig. 1h, i and Supplementary Fig. 2). Taken together, also LH synapses seemingly obey the principle "design rule" of coupling distances previously found at several other synapse types in the *Drosophila* CNS (antennal lobe, calyx, and Kenyon cell synapses)[12,21] as well as at the glutamatergic NMJ synapses[20].

**Both Unc13 isoforms, A and B, contribute to synaptic transmission from ePNs.** To reiterate, Unc13A and Unc13B are both present at the AZs of PNs, though with different nanoscopic spacing. We asked whether these release and priming factors might be functionally redundant, or whether they possibly might transmit distinct aspects of olfactory sensory information. Our previous electrophysiological studies showed that Unc13A encodes a phasic release component when measured from cells postsynaptic to olfactory receptor neurons (ORNs) and MB Kenyon cells. Unc13B, instead, is likely responsible for a slower but sustained release component[12,20,21].

Thus, we knocked down *unc13A* (*unc13A*-knockdown), *unc13B* (*unc13B*-knockdown), or both isoforms (double-knockdown) using isoform-specific RNA interference (RNAi) lines[11] in excitatory PNs driven by GH146-Gal4. Immunostaining against Unc13A and Unc13B revealed a consistent, isoform-specific but still rather moderate reduction of these proteins in the ePN-targeting region of both calyx and LH (Fig. 2a–c).

To assess the efficiency of knockdown, Unc13A and Unc13B intensities were quantified in calyx (Fig. 2b) and lateral horn (Fig. 2c). Importantly, knockdown of either isoform did not affect the level of the respective other isoforms. Moreover, simultaneous double knockdown reduced both isoforms exactly as efficiently as after a single knockdown (Fig. 2b, c).

We also estimated the relative levels of Unc13A versus Unc13B in calyx and lateral horn, by generating a self-made antibody (anti-Unc13$^{C-term}$)[11] detecting the C-terminal part common to both Unc13A and B (Supplementary Fig. 3a, b). We used a pan-neuronal knockdown (using Appl-Gal4)[23] of either isoform to calculate the level of corresponding other isoforms by measuring the remaining Unc13$^{C-term}$ signal. The analysis indicates that Unc13B is somewhat more abundant than Unc13A at both calyx and lateral horn AZs (Supplementary Fig. 3).

We next asked whether the nano-domain organization of either isoform would be changed when the respective other isoform was downregulated. In *unc13A* knockdown and *unc13B* knockdown animals, we measured the localization of the respective other isoforms relative to the BRP ring centers (identifying the center of the AZ and center of Ca$^{2+}$ channel) in calyx ePN boutons (Fig. 2d, e). STED analysis revealed that the nanoscopic localizations are not affected by the reduced physical presence of the respective other isoforms (Fig. 2f). Thus, consistent with their distinct molecular targeting logic, both isoforms[12], and release components can obviously be modulated in an orthogonal fashion.

**Unc13A specifically promotes a fast, phasic release component at ePN output synapses.** Before entering into the behavioral analysis, we sought to test whether indeed Unc13A and Unc13B at PN synapses would play distinct roles in the synaptic transfer of odor-driven, action potential encoded information. We tested for physiological consequences after *unc13A* or *unc13B* knockdown by measuring postsynaptic odor-induced Ca$^{2+}$ transients in Kenyon cell dendrites. As the lateral horn is considered to have a major role in generating innate responses to olfactory cues, measuring ePN release characteristics at lateral horn neurons (LHNs) obviously would have been interesting as well. However, the lack of specific driver lines covering a broad range of LHNs[24] as well as the absence of "morphological landmarks" (such as the microglomerulus organization in the calyx[19]) to identify direct ePN bouton::LHN contacts, unfortunately, precludes reliable measurements of synaptic kinetics here as yet.

In animals with Unc13A or Unc13B downregulation in the presynaptic PN boutons, odor-induced postsynaptic Ca$^{2+}$ responses were imaged with 33 Hz sampling speed in the calycal Kenyon cell dendrites expressing GCaMP3 sensor fused to the postsynaptic scaffold protein Homer[25] (Supplementary Fig. 4). Animals were stimulated with 4-methylcyclohexanol and benzaldehyde as aversive stimuli and 2, 3-butanedione as appetitive stimulus (Fig. 3a–o). Notably, amplitudes of postsynaptic Ca$^{2+}$ transients were significantly reduced in animals in which either *unc13A* or *unc13B* was downregulated (Fig. 3b, c, g, h, l, m). Thus, both Unc13A- but also Unc13B-dependent release components significantly and specifically contribute to the signal transduction from ePNs to their postsynaptic partner neurons. Consistently, double knockdown of *unc13A* and *unc13B* lead to a further reduction in the amplitude of postsynaptic Ca$^{2+}$ response when compared with both single knockdowns of either *unc13A* or *unc13B* (Fig. 3b, c, g, h, l, m).

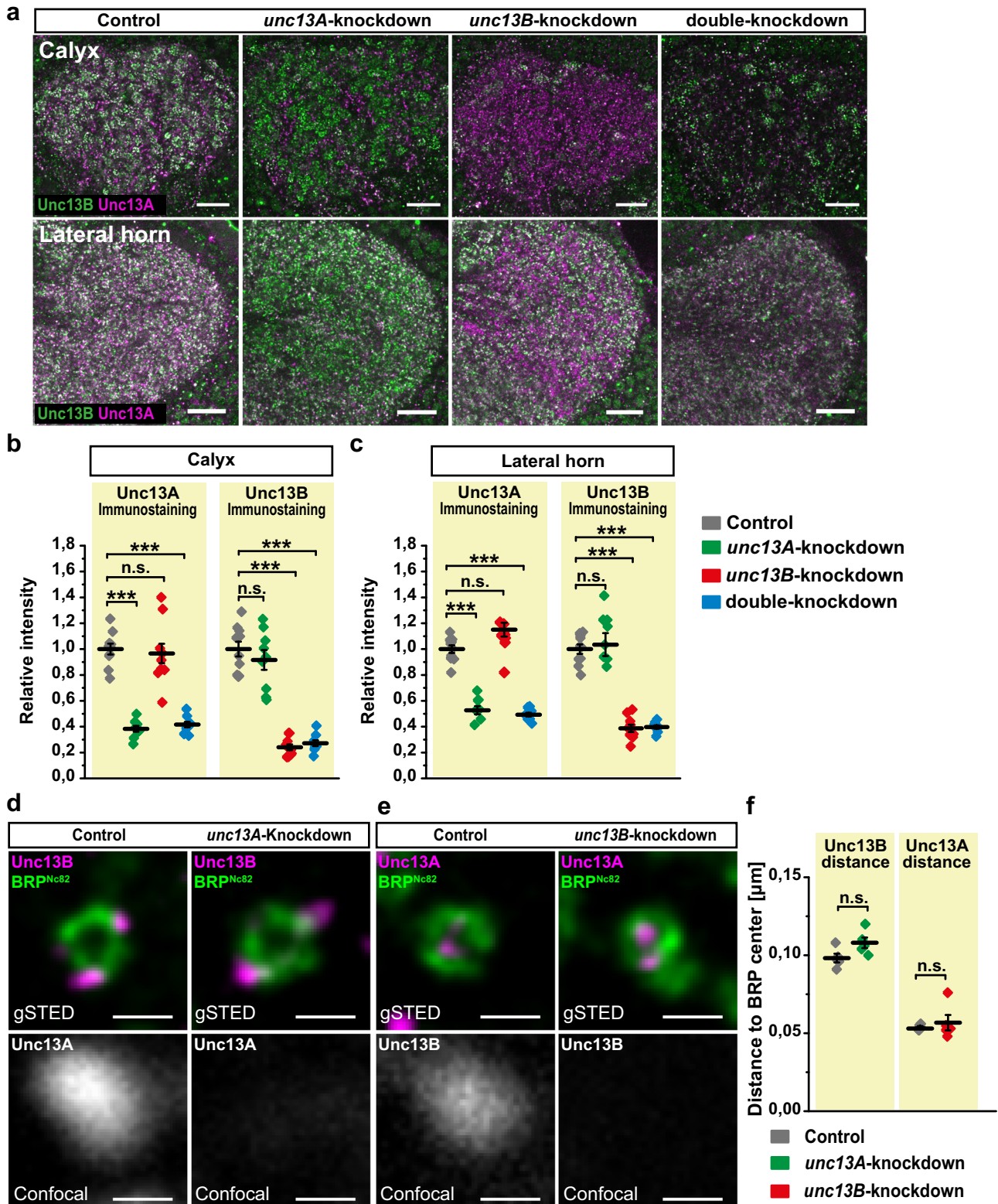

We previously by direct patch clamp recordings of PNs showed that at ORN:PN synapses Unc13A steers fast, phasic release. Unfortunately, the electrical properties of the PN target neurons in calyx and LH as yet preclude electrophysiological recording of mere synaptic signals in sufficient quality[26]. Therefore, in order to characterize the timing of the Unc13A and Unc13B release components at PN output synapses, we analyzed the kinetics of the $Ca^{2+}$ signals. As expected, both Unc13A and Unc13B to a

similar degree contributed to the overall signal transfer (Max $\Delta F/F_0$ (%), Fig. 3b, g, l). Notably, analyzing the kinetics of odor onset across several odors, we observed significantly longer latencies to peak in *unc13A* knockdown and double-knockdown animals, whereas *unc13B* knockdown did not influence the timing of signal onset compared to controls (Fig. 3d, e, i, j, n, o). No correlation between release kinetics (latencies) and $Ca^{2+}$ signal amplitudes (Max $\Delta F/F_0$) for each responsive microglomerulus

**Fig. 2 Specific knockdown of unc13A and unc13B in ePNs. a** Confocal images from the calyx (upper) and LH (lower) regions in animals expressing UAS: unc13A-RNAi (*unc13A*-knockdown) or UAS: unc13B-RNAi (*unc13B*-knockdown) or both (double-knockdown) under control of GH146-Gal4, compared with heterozygous driver line, GH146 (control), immunostained against Unc13A (magenta) and Unc13B (green). Scale bars: 10 μm. **b, c** Expression level of Unc13A and Unc13B proteins in calycal boutons (**b**) and lateral horn (**c**) regions in animals expressing UAS: unc13A-RNAi (*unc13A*-knockdown) or UAS: unc13B-RNAi (*unc13B*-knockdown) or both (double-knockdown) under control of GH146-Gal4, compared with heterozygous driver line, GH146 (control). The values show a significant reduction of Unc13A in *unc13A*-knockdown group (green plot) compared with the control group (gray plot) while no change was observed for Unc13B level. The level of Unc13B significantly reduced in *unc13B*-knockdown group (red plot) compared with the control group (gray plot) while no change was observed for Unc13A level. The level of both Unc13A and Unc13B significantly reduced in the double-knockdown group (blue plot) compared with the control group (gray plot). Control group: $n = 10$, *unc13A*-knockdown: $n = 9$, *unc13B*-knockdown: $n = 10$, double-knockdown: $n = 9$ (**b**). Control group: $n = 10$, *unc13A*-knockdown: $n = 8$, *unc13B*-knockdown: $n = 10$, double-knockdown: $n = 9$ (**c**). *P* values were calculated via one-way ANOVA with the Bonferroni multiple comparison post hoc test (n.s., not significant ($p > 0.05$), ***$p \leq 0.001$). **d–e** Confocal (lower) images of Unc13A (**d**) or Unc13B (**e**) immunostaining and time-gated STED (gSTED; upper) images of BRP [Nc82] (green) and Unc13B (**d**) or Unc13A (**e**) (magenta) at MB calyx. Scale bars: 200 nm. **f** The Unc13B nano-distribution relative to the BRP ring center is unchanged in *unc13A* knockdown (green plot) group in comparison with the control gray (gray plot). Similarly, *unc13B* knockdown (red plot) does not significantly alter the distance between Unc13A and the BRP ring center. *P* values were calculated via two-tailed Mann–Whitney *U* test (n.s., not significant ($p > 0.05$), $n = 5$ brains, 24–30 synapses per brain). Values represent mean ± SEM. Source data including the exact sample sizes and the *p* values are provided as a Source Data file. See also Supplementary Fig. 3.

was observed. Thus, kinetics of the $Ca^{2+}$ signals are evidently not systematically influenced by local odor-driven signal intensities (Supplementary Fig. 5a–c). In addition, in each experimental group, the release kinetics and $Ca^{2+}$ signal amplitudes measured at single microglomerulus level were normally distributed (Supplementary Fig. 5d–i). Our analysis thus suggests that there are no major disparities between different types of KCs here. Taken together, our data provide direct evidence that also at ePN output synapses Unc13A specifically promotes a fast, transient component of SV-release but Unc13B a more tonic release component. Thus, consistent with the specific AZ "nano-architectures" of Unc13A and B being highly similar across synapse types[12,20] (Fig. 1g–i), this specific filtering characteristics can be assumed to be present throughout the *Drosophila* brain.

**Unc13A is needed at ePN output synapses to detect both aversive and appetitive odors**. We next sought to investigate a potentially distinct behavioral role of these two different release components by measuring smell responses in a T-maze assay. With the aim of disentangling potential specific role of Unc13A and Unc13B in olfactory coding, flies with either *unc13A* or *unc13B* knockdown in ePNs were tested (Fig. 4a, b). After *unc13A* knockdown, a significant reduction in olfactory responses toward both aversive and appetitive odors was observed. Downregulation of *unc13B* in ePNs resulted in a significant decrease in the aversion responses toward naturally aversive odors. Surprisingly, however, *unc13B* knockdown showed significantly increased attraction toward appetitive odors (Fig. 4b), in clear contrast to the effects of *unc13A* knockdown. Notably, increasing odor concentrations could restore odor avoidance in both *unc13A* and equally *unc13B* knockdown flies (Supplementary Fig. 6). As described previously[27], expression of tetanus toxin[28] in ePNs using GH146-Gal4 also leads to a significant decrease in olfactory response in a lower odor concentration range[27].

The expression domain of GH146-Gal4 is not absolutely exclusive for ePNs but also drives expression in the GABAergic anterior paired lateral (APL) neuron, with a single cell per hemisphere[29]. Thus, to test for a possible contribution of APL in the GH146-Gal4 driven *unc13A* or *B* knockdown phenotypes, we knocked down *unc13A* or *unc13B* exclusively in APL using VT43924-Gal4[30]. However, we did not observe any significant change in olfactory response in these animals in comparison with the control groups (Supplementary Fig. 7), excluding a major role of this neuron type in our analysis and findings.

Our previous work[12,20] had shown that the AZ amounts of Unc13A and B, also across neuron types, do not scale with their sheer expression levels but are rather defined via the accessibility

of their cognate AZ scaffold protein binding partners, BRP and Syd-1, respectively. Consistently, overexpression of *unc13A-GFP* or *unc13B-GFP* in ePNs did not alter olfactory responses (Supplementary Fig. 8). An additional experiment was based on the observation that the number of Unc13-binding slots in the AZ scaffold is limited. Here, we overexpressed *unc13A*-N-term-GFP fragment in the ePNs. This fragment we previously showed to specifically interfere with Unc13A-dependent release in a dominant negative fashion at larval neuromuscular synapses (NMJs)[11]. The N-term-GFP fragment we here found to interfere with the AZ recruitment of endogenous Unc13A by competitive binding to the AZ scaffold. In accordance with a previous study at NMJs[11], STED analysis revealed that the N-term-GFP fragment expressed in projection neurons clustered at a similar "nano-location" as the endogenous "full-length" Unc13A (58.4 ± 1.2 nm from the BRP ring center). Indeed, overexpression of the *unc13A*-N-term-GFP in the ePNs provoked the same behavioral deficit as *unc13A* knockdown (Supplementary Fig. 9).

We then downregulated both *unc13A* and *unc13B* together in ePNs (Fig. 4c). After double knockdown, behavioral response towards both aversive but also appetitive odors was reduced, creating a situation highly reminiscent of the *unc13A* single knockdown (Fig. 4a, b). Thus, Unc13A is seemingly "epistatic" over Unc13B here, and crucial for "odor recognition as such".

So far, we used in this study "chronic" manipulations starting during development already. We went on by performing *unc13* isoform-specific knockdowns acutely by restricting knockdown to an adult, post-eclosion stage using the tub: Gal80[ts] approach[31]. Animals were kept at 20 °C during development, and gene knockdown was induced after eclosion for 10 days at 29 °C. In order to first test the efficiency of RNAi-mediated knockdown during adult stages, terminals of ePNs (covered by GH146) were identified by expressing a membrane-associated RFP in calyx and LH synapses stained for Unc13A and Unc13B (Supplementary Fig. 10a). Furthermore, to assess whether the knockdown of the respective Unc13 isoform affected either the morphology or the number of synaptic boutons, we also stained for presynaptic AZ protein BRP and postsynaptic density marker Drep2 (Supplementary Fig. 10b). Subsequent quantification of Unc13A and Unc13B intensity in $RFP^+$ and $RFP^-$ boutons in the calyx and the LH regions revealed a significant decline in Unc13A or Unc13B levels in animals expressing the respective RNAi. Importantly, these downregulations remained restricted to the $RFP^+$ boutons but neighboring $RFP^-$ boutons remained unaffected (Supplementary Fig. 10c, d), proof of a truly cell autonomous effect of our knockdown. Moreover, quantification of BRP and Drep2 revealed that the levels of these synaptic proteins remain largely unaffected

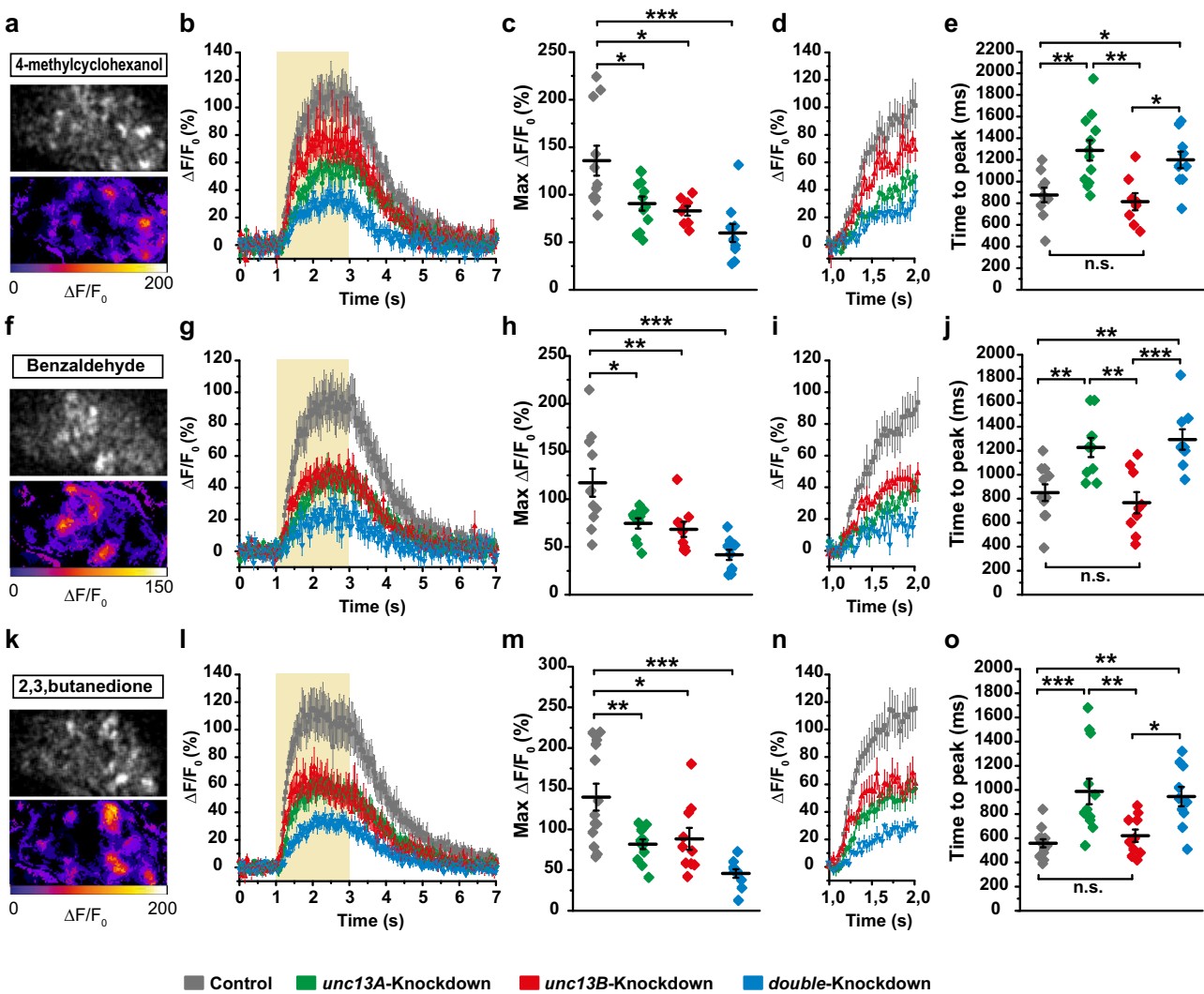

**Fig. 3 Both, Unc13A and Unc13B, specifically contribute to synaptic transmission from ePN output synapses. a–o** In vivo calcium imaging of homer-GCaMP3 using two-photon microscopy at postsynaptic terminals of MB Kenyon cells in the calyx upon odor stimulation, 4-methylcyclohexanol (**a–e**), benzaldehyde (**f–j**) and 2, 3-butanedione (**k–o**). **a, f, k** Representative false color-coded two-photon images of odor-induced fluorescence changes of GCaMP3 in MB calyx (mb247: homer-GCaMP3) at the dendritic terminals of Kenyon cells. **b, g, l** Odor-induced fluorescence change ($\Delta F/F_0$ %) of homer-GCaMP3 in animals expressing UAS: unc13A-RNAi (*unc13A*-knockdown, green traces) or UAS: unc13B-RNAi (*unc13B*-knockdown, red traces) or both (double-knockdown, blue traces) under control of GH146-Gal4 was compared with the control group (GH146-Gal4/+; mb247: homer-GCaMP3/+, gray traces). **c, h, m** Maximum fluorescence changes of GCaMP3 (Max $\Delta F/F_0$ %) upon odor stimulation. The Max $\Delta F/F_0$ (%) in *unc13A*-knockdown (green), *unc13B*-knockdown (red) and double-knockdown (blue) shows a significant reduction compared to the control group (gray). **d, i, n** The $\Delta F/F_0$ (%) traces of the first second of odor stimulation for 4-methylcyclohexanol (**d**) benzaldehyde (**i**), and 2, 3-butanedione (**n**). **e, j, o** Time to maximum fluorescence changes of GCaMP3 (time to peak) upon odor stimulation. The time to peak (ms) is significantly increased in *unc13A*-knockdown (green) and double-knockdown (blue) groups compared with the control group while no change was observed for *unc13B*-knockdown group (red). Values represent mean ± SEM. Control group: $n = 11$, *unc13A*-knockdown: $n = 12$, *unc13B*-knockdown: $n = 8$, double-knockdown: $n = 10$ (**b–e**). Control group: $n = 11$, *unc13A*-knockdown: $n = 10$, *unc13B*-knockdown: $n = 9$, double-knockdown: $n = 9$ (**g–j**). Control group: $n = 13$, *unc13A*-knockdown: $n = 12$, *unc13B*-knockdown: $n = 10$, double-knockdown: $n = 10$ (**l–o**). All *p* values were calculated via one-way ANOVA with the Bonferroni multiple comparison post hoc test (n.s., not significant ($p > 0.05$), *$p \leq 0.05$, **$p \leq 0.01$, ***$p \leq 0.001$). Source data including the exact sample sizes and the *p* values are provided as a Source Data file. See also Supplementary Fig. 4 and 5.

upon Unc13A or Unc13B downregulation (Supplementary Fig. 10e, f), suggesting the absence of gross synaptic organizational changes here.

Importantly, temporally restricted knockdown of the respective Unc13 proteins in the adult stage caused physiological deficits, as determined by Ca²⁺ imaging to a similar degree than after knockdown throughout the entire development (Supplementary Fig. 11).

Based on these findings, we continued to analyze whether this temporally restricted knockdown of the respective Unc13 proteins

in the adult stage would cause similar behavioral deficits like after chronic "developmental" manipulation. Testing for behavioral olfactory responses in the T-maze assay after adult stage-specific knockdown, we observed the same behavioral alterations. For *unc13A*, adult-specific ePN knockdown significantly reduced the reaction scores toward both aversive and appetitive odors. By contrast, adult ePN knockdown of *unc13B* provoked a decreased repulsion toward aversive odors but an increased attraction toward appetitive odors (Fig. 4d, e). Thus, we can conclude that the specific behavioral changes after reduction of Unc13A and

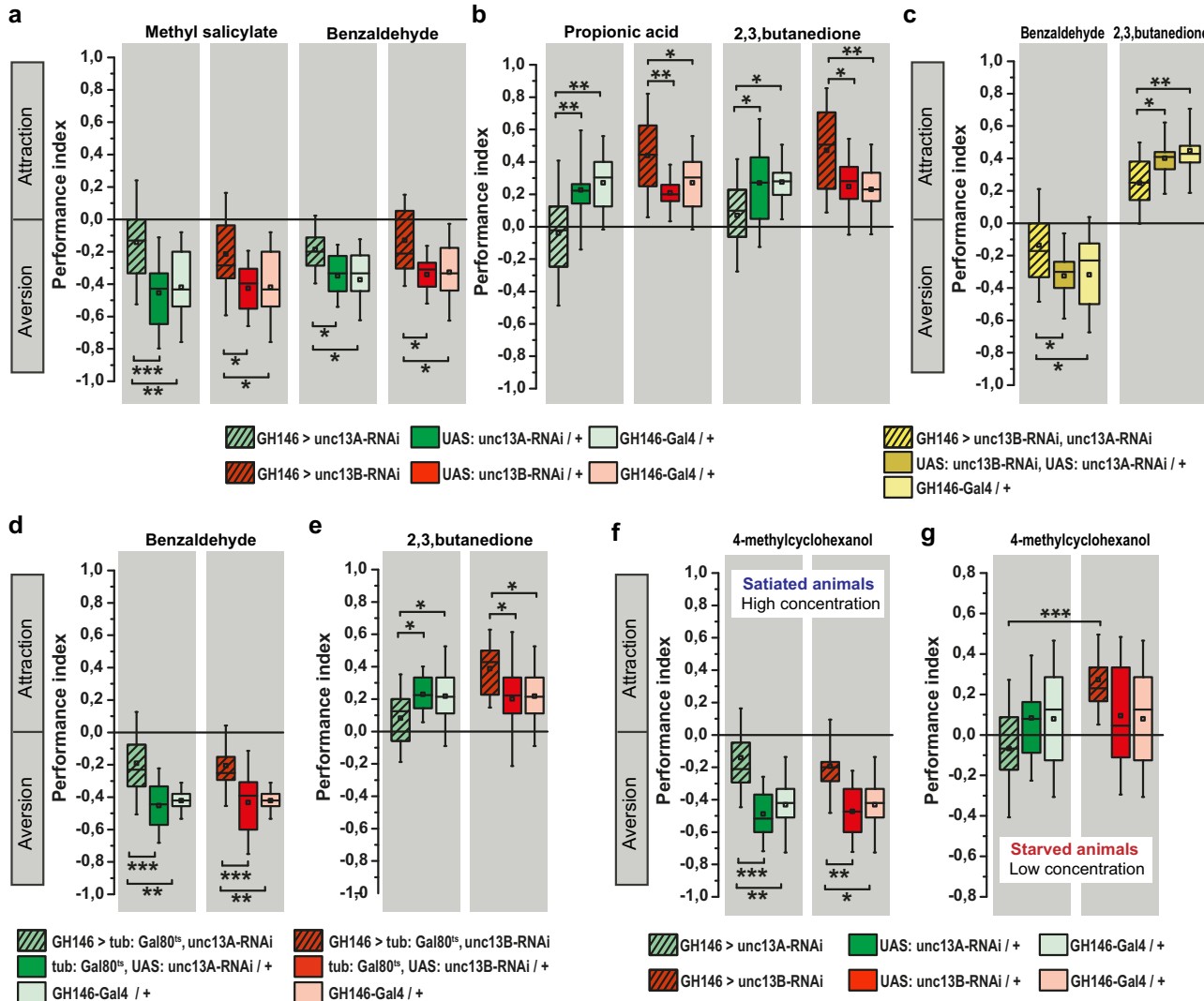

**Fig. 4 Distinct behavioral consequences of Unc13A and Unc13B knockdown in ePNs. a** Knockdown of *unc13A* or *unc13B* under control of GH146-Gal4 significantly reduces the performance index of animals toward aversive odors, methyl salicylate (*n* = 14–17) and benzaldehyde (*n* = 10–12). **b** Performance index of animals with *unc13A* knockdown in ePNs under control of GH146-Gal4 shows significant reduction toward appetitive odors, propionic acid (*n* = 16–17) and 2, 3-butanedione (*n* = 15–17), compared with their genetic controls. Downregulation of *unc13B* under control of GH146-Gal4 leads to increased attraction toward appetitive odors, propionic acid (*n* = 17), and 2, 3-butanedione (*n* = 16–17). **c** Performance index of animals with simultaneous *unc13A* and *unc13B* knockdown in ePNs under control of GH146-Gal4 shows a significant reduction toward appetitive odor, 2, 3-butanedione (*n* = 14–17), and aversive odor, benzaldehyde (*n* = 17–22), compared with the genetic control. **d** Knockdown of *unc13A* or *unc13B* under control of GH146-Gal4 during adult stage using tub: Gal80^ts significantly reduces the performance index of animals toward aversive odor benzaldehyde (*n* = 12–17). **e** Performance index of animals with *unc13A* knockdown in ePNs under control of GH146-Gal4 during adult stage shows significant reduction toward appetitive odor, 2, 3-butanedione (*n* = 18–25) compared with their genetic controls. Knockdown of *unc13B* under control of GH146-Gal4 during adult stage leads to an increased attraction toward appetitive odor, 2, 3-butanedione (*n* = 23–25). **f** Knockdown of *unc13A* or *unc13B* under control of GH146-Gal4 significantly reduces the performance index of satiated animals toward high concentration of aversive odor 4-methylcyclohexanol (*n* = 10–12). **g** Knockdown of *unc13A* under control of GH146-Gal4 significantly reduces the performance index of starved animals toward low concentration of aversive odor 4-methylcyclohexanol (*n* = 17–23) in comparison to the starved animals with *unc13B* knockdown in ePNs. All *p* values were calculated via one-way ANOVA with the Bonferroni multiple comparison post hoc test (\**p* ≤ 0.05, \*\**p* ≤ 0.01, \*\*\**p* ≤ 0.001). Box plots indicate means (black squares), medians (middle lines), 25th and 75th percentile ranges (boxes), and SD ranges (whiskers). Source data including the exact sample sizes and the p values are provided as a Source Data file. See also Supplementary Fig. 6, 7, 8, 9, 10, 11.

Unc13B levels are most likely directly attributable to the specific, distinct roles that these two release components play for synaptic function: phasic versus tonic release that convey distinct types of olfactory information.

**Unc13B release at ePN synapses orchestrates a valence independent "go-away signal".** We further analyzed the functional relation of Unc13A and Unc13B-mediated behavioral responses. So far, our results could be interpreted as a specific difference between Unc13A and Unc13B only concerning appetitive odors (*unc13A* knockdown reducing attraction, *unc13B* knockdown increasing attraction), as for aversive odors knockdown of *unc13B* resulted in reduced aversion (a similar scenario as *unc13A* knockdown). As an alternative interpretation, the Unc13A release component might play an indispensable role in mediating the

response to either kind of odors, whereas the Unc13B ePN transmission component could transmit a generic, valence independent "go-away" signal on odor interpretation of either kind, which would behave additive to the Unc13A mediated response. To discriminate between these possibilities, we screened for an odor which would be aversive under standard conditions but changing valence in a context-dependent manner. Under standard conditions (flies satiated, medium to higher odor concentration), 4-methylcyclohexanol provoked an aversive response, which was significantly and similarly reduced after ePN knockdown of either *unc13A* or *unc13B* (Fig. 4f). However, presenting the same odor but in lower concentration and on starved animals in controls resulted in attraction. Notably, although *unc13A* knockdown again reduced the olfactory response (to a value indistinguishable from the per se aversive conditions), *unc13B* knockdown severely increased the attractive response over control levels (Fig. 4g), resulting in a significant difference of odor responses when comparing *unc13A* with *unc13B* ePN knockdown (compare unc13A-RNAi with Unc13B in Fig. 4b).

**Interfering with nanoscopic patterning of ePN release sites eliminates smell response.** To directly investigate the relevance of the distinct presynaptic nano-organization of Unc13 defining release sites in the decoding of behaviorally relevant information, we overexpressed the C-terminal region of the *unc13A* (*unc13A*-C-term-GFP). Thus, we expressed an Unc13A-C-terminal fragment in ePNs (Fig. 5a, b), whose effects we had studied in detail at NMJ synapses rendering themselves for biophysical analysis[11]. At NMJ AZs, expressing this C-terminal fragment generated release sites at atypical locations, means it interfered with proper nanoscale organization. This in turn resulted in severe changes of

the release time course[11], showing that specific AZ localization via the N-terminus prevents ectopic, temporally imprecise release. When expressing in ePNs, as expected from our NMJ work, we observed that the Unc13A-C-terminal fragment efficiently reached the synaptic terminals, but instead of strongly enriching at the AZs (labeled with staining against endogenous full-length Unc13A) it formed AZ ectopic clusters at the terminal inner surface (Fig. 5b). Most importantly, the Unc13A-C-terminal fragment expression provoked a near elimination of the smell response for both aversive and appetitive odors (Fig. 5c, d). Expressing the C-terminal fragment together with Unc13A knockdown resulted in a similarly drastic phenotype. These data suggest that indeed the nano-organization of Unc13 isoforms is indispensable for patterned synaptic transfer of olfactory information and consequently smell behavior.

**Unc13B-mediated transmission components of ePNs and iPNs operate antagonistically.** We finally asked in which neuropile the signal integration of Unc13A- and B-specific transmission might take place. Unfortunately, up to now, genetic analysis does not allow to specifically distinctly manipulate the different synapse types formed by a single neuron type. As ePNs form presynapses in the LH, MB calyx, and also antennal lobe, we a priori could not assign the behavioral differences evoked by Unc13A versus B knockdown to these subtypes.

Notably, current concepts propose that LH and MB predominantly mediate innate and learned behaviors, respectively. The fact that we had observed differences in innate behavior favored the ePN LH input as most relevant here. Indeed, when we acutely suppressed MB neuron output (using expression of the temperature sensitive dynamin allele, *shibire* (Shi^ts)[32,33]),

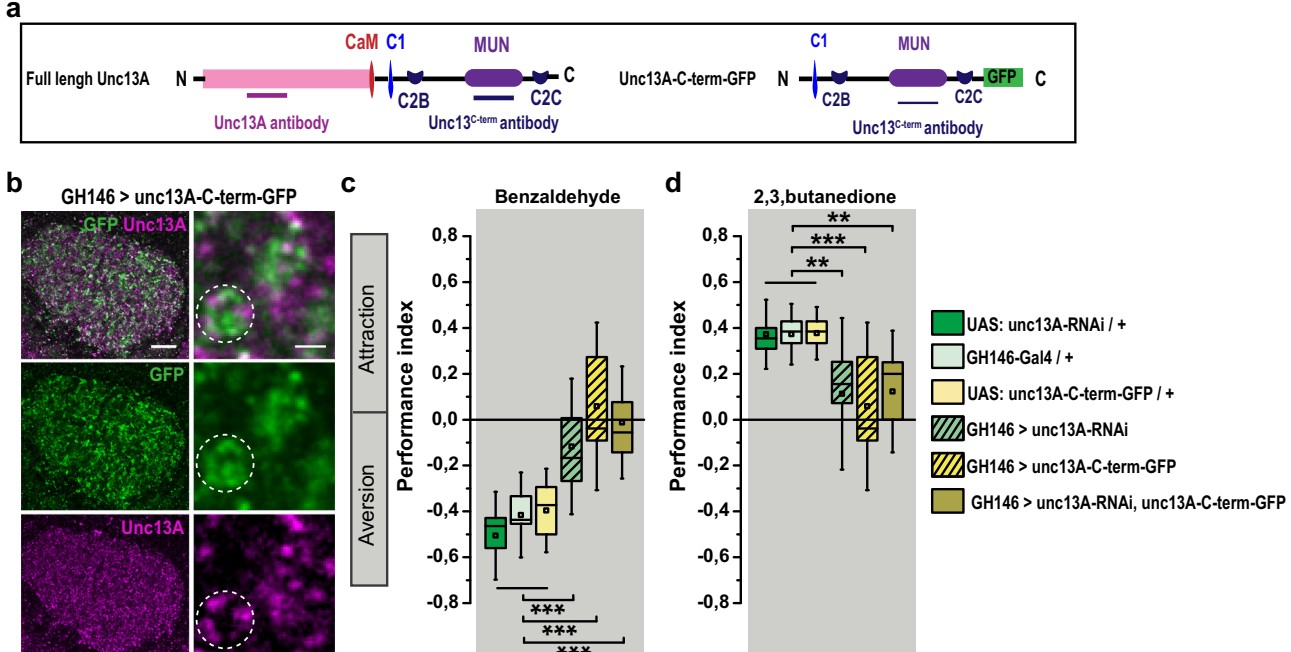

**Fig. 5 Olfactory impairment in animals overexpressing unc13A-C-term-GFP in ePNs. a** Schematics of full-length Unc13A and Unc13A-C-term-GFP constructs indicating CaM, calmodulin; C1, C2B, C2C, and the MUN domain. **b** Confocal images from the calyx region in animals expressing UAS: *unc13A*-C-term-GFP in ePNs under control of GH146-Gal4, immunostained against Unc13A (magenta) and GFP (green) Scale bars: 10 µm, in inset: 500 nm. The white circles indicate the localization of C-term-GFP fragment (green) between endogenous Unc13A clusters ($n = 4$, experiment was performed once). **c, d** Performance index of animals with *unc13A* knockdown in ePNs, overexpression of *unc13A*-C-term-GFP in ePNs or both show significant reduction toward aversive (**c**) and appetitive (**d**) odors compared with their genetic controls ($n = 12–14$). All $p$ values were calculated via one-way ANOVA with the Bonferroni multiple comparison post hoc test (**$p \leq 0.01$, ***$p \leq 0.001$). Box plots indicate means (black squares), medians (middle lines), 25th and 75th percentile ranges (boxes), and SD ranges (whiskers). Source data including the exact sample sizes and the $p$ values are provided as a Source Data file.

responses to appetitive odors were not affected, again favoring the LH over the MB for mediating the *unc13* isoform knockdown behavioral effects (Supplementary Fig. 12). However, we would like to emphasize that for aversive odors, the same experiment did indeed change responses, suggesting a role of the MB for innate behavior towards aversive odors.

Previous studies showed that the GABAergic input from iPNs regulates the odor input into LH neurons as well[34–37]. Interestingly, iPN release seems to encode positive hedonic valence in LH as silencing of GABA release from iPNs (using GAD-RNAi) was shown to provoke a shift toward aversion in innate smell responses[36]. Thus, concerning innate olfactory responses, a generic reduction of iPN release results in exactly the opposite behavioral outcome as we observed after reducing Unc13B component in ePNs (Fig. 4a, b, d, e).

We, therefore, tested how iPN-specific *unc13A* or *unc13B* knockdown would intersect with ePN *unc13B* knockdown. Interestingly, single knockdown of either *unc13A* or *unc13B* in only iPNs did not result in significant changes of olfactory response (Fig. 6a, b). However, co-knockdown of *unc13B* in iPNs and ePNs together did abolish the ePN-only knockdown provoked behavioral reprogramming towards attraction (Fig. 6c).

In mice, the two isoforms of Munc13, Munc13-1 and Munc13-2, operate redundantly in inhibitory hippocampal neurons[38,39]. We asked whether Unc13A and Unc13B also acted redundantly in iPNs. Therefore, we generated a new transgenic line (*QUAS: unc13B-RNAi*), enabling us to specifically and exclusively down-regulate *unc13B* and *unc13A* in ePNs and iPNs, respectively, by combining the two binary expression systems, UAS/Gal4 and QUAS/QF. Efficient downregulation by the QUAS/QF system was validated by immunostaining (Supplementary Fig. 13) and by reproduction of the ePN behavioral effect (Fig. 6d). Different from *unc13B* iPN knockdown, knockdown of *unc13A* in iPNs did not restore the behavioral consequence of *unc13B* knockdown in ePNs (Fig. 6e). In addition, using the same binary expression systems to downregulate *unc13B* in ePNs via QUAS/QF and iPNs via UAS/Gal4 leads to abolishment of the behavioral phenotype observed in *unc13B* knockdown in ePNs (Fig. 6f, compare to Fig. 6c).

As the iPNs directly target the LH but bypass the MB, our results point towards the lateral horn as the place where Unc13A/B exert differential effects on inducing innate olfactory behavior, and imply a quasi-antagonistic computation between the Unc13B-dependent transmission components of ePNs and iPNs.

## Discussion

Synapses are functionally highly heterogeneous. Synaptic STP is particularly diversified and considered critical for routing and encoding sensory information within neuronal circuits[1,40]. Consequently, STP has the potential to contribute to the temporal coding of multisensory integration and extraction of specific sensory features. However, direct genetic manipulations to test for the role of STP in a behaviorally relevant context have remained scarce. We here start from the finding that in the *Drosophila* brain-specific STP features of release components could be assigned to two Unc13 species, Unc13A and Unc13B[12], and that thus distinct STP components became genetically accessible in this model for the study of olfactory behavior. Our previous analysis targeting the first relay synapse of the *Drosophila* olfactory system, connecting ORNs with projection neurons, had found that Unc13A promoted a high probability and fast depressing phasic release component, while Unc13B supported a slower tonic release component[12]. Distinct "coupling distances" of release components obviously are associated with distinct STP behavior. Notably, both release components, Unc13A and B, were

co-existing in individual AZs, likely being directly responsible for the fact that this synapse exhibits short-term depression (Unc13A-dependent component), but still transmits broadband signals[41]. Indeed, mixing of different release components with specific STP features seems to underlie more complex release features, as is obviously the case at the ORN::PN synapse.

In this study, we now show that also within individual AZs of projection neurons, Unc13A and Unc13B clusters directly co-exist, with their nanoscopic spacing relative to central $Ca^{2+}$ channels being very similar as found at ORN-derived[12] and Kenyon cell-derived[21] AZs. This should similarly structure their STP features and allow for the extraction of different frequency components from the time-encoded sensory information. Unfortunately, electrophysiological recordings and analysis from the ePN postsynaptic partners in sufficient quality (as for ORN-PN synapses) could not be directly demonstrated due to space clamp problems. Yet, the fact that the genetic attenuation of both components resulted in clearly qualitatively different behavioral phenotypes (Figs. 4 and 6) argues in favor of the Unc13A and B indeed playing different functional roles at PN output synapses as well. Analyzing how the complex and parallel temporal coding of olfactory information and these synaptic filter functions are co-adapted should be a warranting subject of future research[42].

Our data suggest that the ePN temporal code contains distinct components specifically filtered by the two release components at their output synapses, Unc13A and Unc13B. Indeed, projection neurons have been shown to support a broad distribution of action potential frequencies in response to odors[35,43]. Seemingly, under the conditions chosen in this study, Unc13B-related signaling encodes an avoidance or "go-away signal" in response to the odor stimulation. Attenuating the Unc13B-mediated transmission signal led to more attraction towards odors in our smell assay. It thus appears that when reacting to appetitive cues, the Unc13A transmission-mediated signal seemingly has to overcome an Unc13B "counter force".

It is tempting to speculate that Unc13B-related signaling might be an "information channel" to reflect the internal state of animals. Neuromodulation is well known to play a crucial role in adopting animals to changing circumstances. Recently, a unique type of serotoninergic neurons (CSDn) was found to change the responsivity to the synaptic input by using opposing coding with different functions in AL and LH. In this way, the information about stimulus features, odor intensity, and identity can be segregated by neuromodulation[44]. Based on our data we propose the existence of a layer of encoding for context-dependent olfactory behavioral output in the same type of second-order neurons (ePNs). Notably, hunger increases the response in the DM1 glomerulus through the sNPF neuropeptide[45] but decreases the DM5 glomerulus response by tachykinin signaling[46]. Our data now open the possibility of phasic and tonic release components to be differentially controlled or read out in the behavior relevant downstream decoding of temporal information.

The parallel olfactory pathways and interaction between ePNs and iPNs have been suggested to be essential for appropriate behavioral responses in courtship and aggression[37]. In addition, it has been shown that reduction in GABA release from iPNs shifts the behavior towards aversion[36], whereas our data show that *unc13B* knockdown in ePNs leads to a behavioral shift towards more attraction, and that simultaneous downregulation of *unc13B* in both ePNs and iPNs cancels the effects out. Since blocking synaptic output from the MB did not interfere with sensing appetitive odors, we favor the idea that the LH is the region in which ePNs and iPNs and their release components functionally interact. Previous anatomical and functional evidence did not suggest direct synaptic interactions between ePNs and iPNs[34,37]. Therefore, we suggest the possibility of a convergence of the

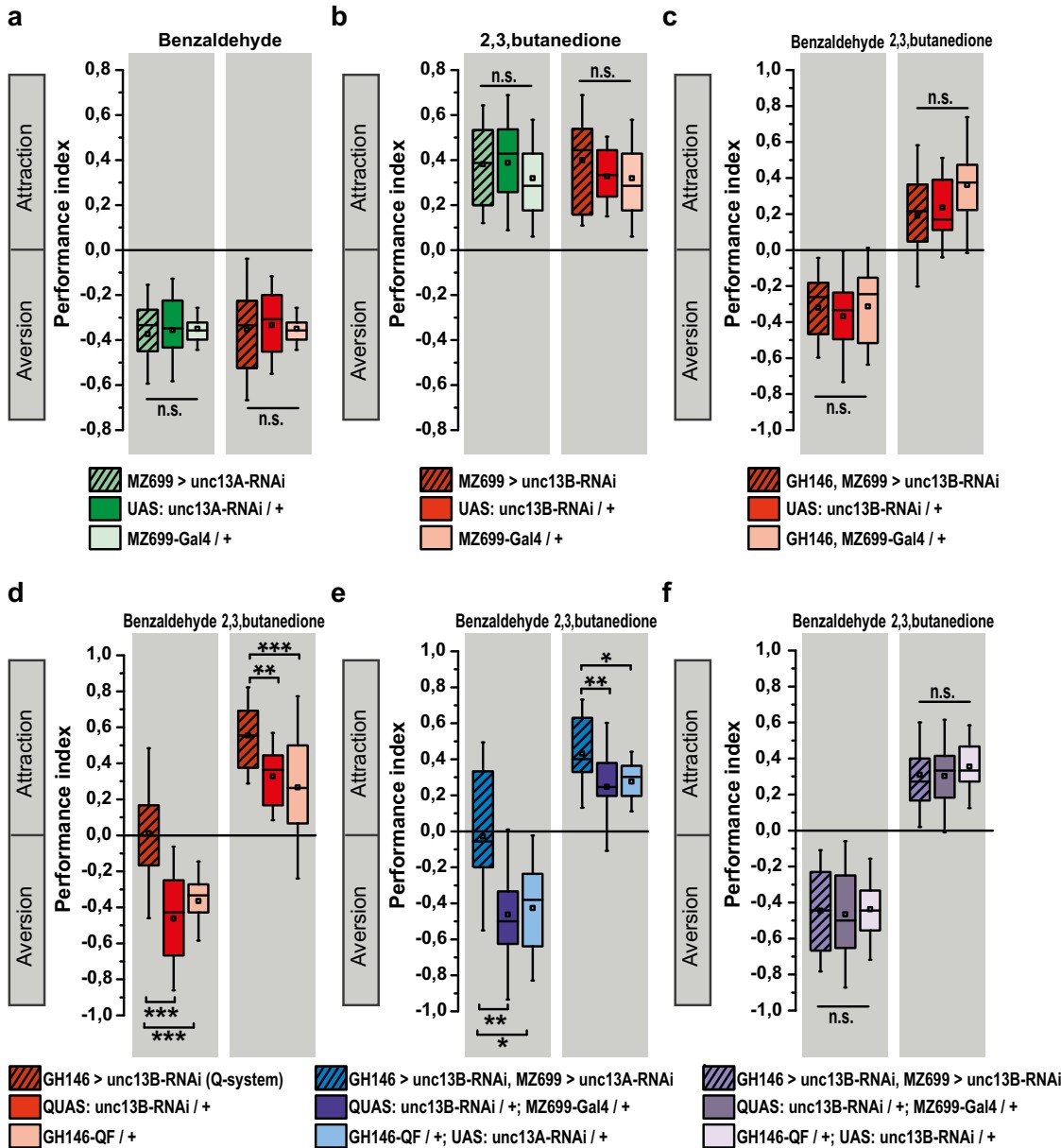

**Fig. 6 Unc13B-mediated transmission components in ePNs and iPNs operate antagonistically. a**, **b** *unc13A* knockdown or *unc13B* knockdown in iPNs using MZ699-Gal4 driver line has no significant effect in performance index of animals toward aversive (benzaldehyde $n = 12–17$) and appetitive (2, 3-butanedione, $n = 13–20$) odors. **c** Knockdown of *unc13B* in both ePNs and iPNs using combined driver lines, GH146-Gal4, MZ699-Gal4, eliminates the behavioral effect of *unc13B* knockdown in ePNs. No significant difference in performance index of animals with ePN-iPN *unc13B* knockdown toward appetitive and aversive odors in comparison to the control groups was observed (benzaldehyde $n = 14–16$, 2, 3-butanedione, $n = 17–18$). **d** Knockdown of *unc13B* in ePNs using QUAS/QF system leads to reduced aversion toward aversive odor (benzaldehyde, $n = 21–23$) and increased attraction toward appetitive odor (2, 3-butanedione, $n = 21–26$). **e** Simultaneous knockdown of *unc13A* in iPNs using UAS/Gal4 system and knockdown of *unc13B* in ePNs using QUAS/QF leads to the significant reduction in aversion toward aversive odor (benzaldehyde, $n = 10–12$) and increased attraction toward appetitive odor (2, 3-butanedione, $n = 22–26$). **f** Knockdown of *unc13B* in iPNs using UAS/Gal4 system and knockdown of *unc13B* in ePNs using QUAS/QF eliminate the behavioral effect of *unc13B* knockdown in ePNs. No significant difference in performance index of animals with ePN-iPN *unc13B* knockdown (using two binary systems) toward appetitive and aversive ($n = 12–14$) odors in comparison with the control groups was observed. All $p$ values were calculated via one-way ANOVA with the Bonferroni multiple comparison post hoc test (n.s., not significant ($p > 0.05$), *$p \leq 0.05$, **$p \leq 0.01$, ***$p \leq 0.001$). Box plots indicate means (black squares), medians (middle lines), 25th and 75th percentile ranges (boxes), and SD ranges (whiskers). Source data including the exact sample sizes and the $p$ values are provided as a Source Data file. See also Supplementary Fig. 12, 13.

GABAergic signaling from iPNs and cholinergic signaling from ePNs to overlapping populations of higher-order LH neurons.

Notably, the *Drosophila* brain is a focus point of large, concerted efforts to use systematic, brain-wide electron microscopic reconstruction to decipher circuit motifs operating in information decoding[47]. We now demonstrated that nanoscopically diversified

release components of different coupling distances co-exist at AZs, which apparently play distinct roles in the physiological and behavioral context. The fact that the ratio between these two release components, Unc13A and B, is highly diversified across synapse types of all parts of the *Drosophila* brain[12], is of clear relevance here. For example, high Unc13B levels might point

towards synapses "seeing" high-frequency information. Nanoscopic molecular synaptic "fingerprints" accessible via super-resolution microscopy should thus be able to complement the in toto ultrastructural reconstruction of the *Drosophila* olfactory system and brain when analyzing information encoding and decoding.

In the future, it will be interesting to investigate how the distinct synaptic filtering functions executed by Unc13A and Unc13B steer the downstream signal computation processes in both, the MB and importantly the lateral horn.

## Methods

**Fly rearing and strains**. *Drosophila melanogaster* stocks were raised on conventional cornmeal-agar medium at 25 °C temperature and 60% humidity and a 12 hr light:12 hr dark cycle. For chronic RNAi experiments, flies were raised at 29 °C for increasing the efficiency of RNAi expression and for acute tub: Gal80[ts] experiments, flies were kept at 20 °C during development, and knockdown was induced after eclosion for 10 days at 29 °C. For Shi[ts] experiment, flies were raised at 20 °C. For starvation, flies (3–4 day old) were starved for 24 hr at 25 °C. The following fly strains were used for the different experiments: GH146-Gal4[15], MZ699-Gal4[16], UAS:unc13A-RNAi[20], UAS:unc13B-RNAi[20], sfCac[GFP,22], UAS:brp-short [GFP,13,14], tub: Gal80[ts] (BDSC # 7019[31]), UAS: myr-RFP (BDSC # 7118), GH146-QF, QUAS: mcd8-GFP (BDSC # 30038), UAS:unc13A-N-term-GFP[11], VT30559-Gal4 (VDRC # v206077), 20xUAS:Shi[ts,33], Appl-Gal4 (BDSC # 32040), VT43924-Gal4 (VDRC # v201194). For generating transgenic flies expressing the postsynaptically localized Ca$^{2+}$ sensor homer- GCaMP3 under control of the MB-specific promoter mb247[48], the DNA sequence of dhomer-GCaMP3[25] was cut from the pUAST vector and inserted into the pCaSpeR-mb247 vector[49] using XbaI and EcoRI restriction enzyme sites. Germline transformation was carried out by BestGene Inc. (CA). For generating transgenic flies expressing *QUAS:unc13B-RNAi*, we used *pWalium20-unc13B*-RNAi vector[12] and the UAS sequence and gypsy insulator were replaced by QUAS sequence from *pQUASp* (addgene # 46162) and gypsy insulator using StuI and BamHI restriction enzymes. The *QUAS* sequence and gypsy insulator were combined together with elongase PCR technique. Sequences of the primers used are listed in Supplementary Table 1. Germline transformation was carried out by BestGene Inc. (CA, USA).

**Olfactory T-Maze assay**. Olfactory performance index was measured in a T-maze apparatus modified from Tully and Quinn[50]. Flies were tested in groups of 30–40 flies in a T-maze and were allowed 1 min to make a decision to go into either arm. Experimentation was carried out within climate-controlled boxes at 25 °C and 70% humidity in the dark. For Shi[ts] experiment the experimental temperature was set to 29 °C. Odor solution (Sigma-Aldrich) at different concentrations diluted in distilled water or paraffin oil were presented in cups of 14 mm diameter. The odors were used in the following concentrations: methyl salicylate (diluted 1:1000 in paraffin oil), benzaldehyde (diluted 1:2000 in paraffin oil, propionic acid (diluted 1:100 in water), 2, 3-butanedione (diluted 1:500 in water), 4-methylcyclohexanol (diluted 1:10000 or 1:300 in paraffin oil). The performance index was calculated as [(number of flies in odor side−number of flies in oil or water side)]/(total number of flies). Statistical analysis was performed using Analysis of variance (ANOVA) and the Bonferroni multiple comparisons post hoc test.

**Immunostaining**. Brains were dissected in Ringer's solution (pH 7.3, 290–310 mOsm) containing 5 mM HEPES-NaOH, 130 mM NaCl, 5 mM KCl, 2 mM MgCl$_2$, 2 mM CaCl$_2$ and 36 mM sucrose, fixed in 4% paraformaldehyde for 2 hr at 4 °C or in methanol for 1 hr 4 °C and washed three times for 10 min each in PBS containing 0.7% Triton X-100 (PBT) at room temperature. Samples were incubated for 2 hr in PBT containing 2% bovine serum albumin (BSA) and 5% normal goat serum. Subsequently, the samples were incubated in the primary antibody diluted in block solution at 4 °C for 48 hr. Samples were washed four times for at least 30 min each in PBT containing 2% BSA (PAT) at room temperature, subsequently incubated with secondary antibody diluted in PAT overnight at 4 °C. Brains were washed at least six times for 30 min each in PBT embedded in Vectashield (Vector Laboratories). For STED microscopy, samples were embedded in Prolong Gold Antifade (Invitrogen) and stored for 24 hr at room temperature followed by 48 hr at 4 °C. For confocal and Airyscan following primary antibodies were used: chicken anti-GFP (Abcam, ab13970, diluted 1:2000), guinea pig anti-Unc13A[20] (diluted 1:300), guinea pig anti-Unc13B[20] (diluted 1:500[20]), rabbit anti-Unc13B[20] (diluted 1:300), rat anti-RFP (Chromotek, 5F8, diluted 1:350), mouse anti-BRP[NC82] (DSHB, diluted 1:30), rabbit anti-Drep2[C−Term,18] (diluted, 1:500) and rabbit anti-Unc13[C−Term,11] (diluted, 1:300). As secondary antibodies, Alexa Fluor 488-coupled goat anti-chicken (Invitrogen A11039, diluted 1:500), Cy3-coupled goat anti-guinea pig (Abcam ab102370, diluted 1:500), Alexa Fluor 633-coupled goat anti-rabbit (Invitrogen A21071, diluted 1:500), Alexa Fluor 647-coupled goat anti-guinea pig (Invitrogen A21450, diluted 1:500), Cy3-coupled goat anti-rat (Invitrogen A10522, diluted 1:500), Alexa Fluor 488-coupled goat anti-guinea pig (Invitrogen A11073, diluted 1:500) and Alexa Fluor 488-coupled goat anti-mouse (Invitrogen A11029, diluted

1:500) were used. For STED microscopy following primary antibodies were used: chicken anti-GFP (Abcam ab13970, diluted 1:1000), guinea pig anti-Unc13A[20] (diluted 1:200), guinea pig anti-Unc13B[20] (diluted 1:200), mouse anti-BRP[NC82] (DSHB, diluted 1:10) and rabbit anti-Drep2[C−Term,18] (diluted, 1:500). As secondary antibodies, Alexa Fluor594-coupled goat anti-guinea pig (Invitrogen 11076, diluted 1:200), STARRED-coupled goat anti-chicken (Abberior STRED-500-500UG, diluted 1:200), ATTO490 LS-coupled goat anti-rabbit (Hypermol 2309, diluted 1:50) or Alexa Fluor 488-coupled goat anti-rabbit (Invitrogen A11034, diluted 1:200), and STARRED-coupled goat anti-mouse (Abberior STRED-1001-500UG, diluted 1:200) were used.

**Confocal imaging and data processing**. Immunostained brain samples were imaged using Leica SP8 confocal microscope equipped with ×20 apochromat Leica objective (NA = 0.75), ×40 Leica apochromat oil-immersion objective (NA = 1.30) and ×63 Leica apochromat oil-immersion objective (NA = 1.40). Alexa Fluor 488 was excited at 488 nm, Cy3 at 552 nm, and Alexa Fluor 633 at 638 nm wavelengths. Samples were scanned using LAS X software (3.5.2.18963) at 1 µm sections in the z direction with a frame average of 4. All images were acquired at 8-bit grayscale. For Airyscan, Zeiss LSM880 inverted microscope equipped with ×63/1.40 oil Plan-Apochromat was used. Alexa Fluor 488 was excited at 488 nm, Cy3 at 561 nm, and Alexa Fluor 633 at 633 nm wavelengths. Samples were scanned using ZEN software (Black 2.3) at 1 µm sections in the z direction with pixel size of 0.04 µm. All images were acquired at 8-bit grayscale. Images were processed using the Fiji software (1.52P)[51] for adjusting brightness, merging of emission channels, and calculating average and maximal intensity projections across the z axis. Mander's overlap coefficient was used to determine the degree of co-localization on 8-bit images using the ImageJ (version 1.52p, NIH) plugin "Coloc 2", with the PSF set to two pixels (Airyscan images) Mander's tM1 overlap coefficients of channel 1 (BRP-short [GFP]) with channel 2 (Unc13 isoform) above auto-threshold of channel 2 were measured in a single ROI (120 × 120 pixels) for a given optical slice within a whole z-stack and four optical slices at different depth per each hemisphere were measured.

**Time-gSTED microscopy and image analysis**. Time-gSTED microscopy was performed using an Abberior Instruments Expert Line STED setup equipped with an inverted IX83 microscope (Olympus), two pulsed STED lasers for depletion at 775 nm (0.98 ns pulse duration, up to 80 MHz repetition rate) and at 595 nm (0.52 ns pulse duration, 40 MHz repetition rate) and pulsed excitation lasers (at 488 nm, 561 nm, and 640 nm). The dyes STARRED, Alexa Fluor594, and ATTO490 LS were depleted with a pulsed STED laser at 775 nm. Alexa Fluor 488 was depleted with a pulsed STED laser at 595 nm. Time gating was set at 750 ps. Fluorescence signals were detected sequentially by avalanche photodiode detectors at appropriate spectral regions. 2D gSTED Images were acquired with a ×100, 1.40 NA oil-immersion objective with a pixel dwell time of 20 µs and 3× lines accumulation or with a pixel dwell time of 2 µs and ×30 lines accumulation at 16bit sampling and a field of view of 10 µm × 10 µm. Lateral pixel size was set to 20 nm. Within each experiment, samples belonging to the same experimental group were acquired with equal settings. For the analysis of Drep2$^+$ and Drep2$^−$ synapses, experiments were repeated three times on different biological replicates. Raw triple-channel gSTED images were processed for Richardson–Lucy deconvolution using the Imspector software (16.1.6477, Germany). The point spread function was automatically computed with a 2D Lorentz function having a full-width half-maximum of 40 nm, based on measurements with 40 nm Crimson beads. Default deconvolution settings were applied.

Deconvolved 8-bit gSTED images were used for quantification of Unc13A/B distances to sfCac spots by line profile estimation of peak-to-peak distances. Line profile measurements of distances between spots were performed in Imagej (NIH). Well-defined planar or side view synapses, having sfCac and Unc13A/B spots on the same focal plane, were manually traced with the line profile tool (thickness 1 pixel/20 nm) and peak intensities across the line were retrieved using the ImageJ Macro (Macro_plot_lineprofile_multicolor from Kees Straatman, University of Leicester, Leicester, UK). Synapses were considered inhibitory and distinguished from excitatory Drep2-positive synapses when the Drep2 signal across the annotated line was below 60 gray values or their Drep2 peak had a consistently lower intensity than the sfCac peak. Intensity values from individual synapses were exported to Excel. Four to eleven line profiles per image were considered for analysis. Local maxima were calculated with the SciPy "argrelmax" function, as described in Brockmann et al.[52], in order to obtain peak intensities for different image channels and peak-to-peak distances. Only highest maxima were selected. Values were then averaged per animal.

Statistic was performed with GraphPad Prism (version 7.03, GraphPad, La Jolla, CA, USA), applying a similar strategy as in Gupta et al.[53]. The non-parametric Kruskal–Wallis test with the Dunn's post hoc test corrected for four multiple comparisons was applied to test differences in mean Unc13A/B-to-sfCac distances between groups. Differences in frequency distribution of Unc13A/B-to-sfCac distances were tested with the two samples Kolmogorov–Smirnov test.

The spatial relation between Unc13A, Unc13B, or Unc13A-N-term-GFP and the BRP[NC82] ring center, was determined on 8-bit deconvolved STED images as described previously[12]. Briefly, six to eight subregions per image, containing individual planar AZs, were selected within 1.06 × 1.06 µm ROIs (53 × 53 pixels) in

ImageJ (version 1.52p, NIH). To identify the exact position of the AZ center, even smaller AZ ROIs were placed on the BRP$^{NC82}$ subimages, tightly surrounding the BRP$^{NC82}$ rings, the lowest intensity pixel value subtracted to the image and the center of mass determined using output values XM and YM of the ImageJ function "Measure". The sameROIs (53 × 53 pixels) were used to retrieve the "X" and "Y" coordinates of local intensity maxima in the second channel via the function "*Find maxima*" (noise = 0 for Unc13A-N-term-GFP; noise = 5 for Unc13A or Unc13B). Subsequently, the Euclidean distances of each AZ protein spots relative to the BRP$^{Nc82}$ ring center of mass, indicating the AZ center, was calculated using Matlab (R2016a, Mathworks, Natick, USA). For each AZ, the observed distances were ranked and the minimum distance selected. Values were averaged per animal.

The displayed gSTED images are deconvolved and rescaled.

**In vivo calcium imaging**. For live imaging transgenic heterozygous female flies (5–6 day old for chronic and 10–11 day old for acute experiment) were used. Imaging was performed using two-photon microscope (Leica) equipped with a ×20 water–immersion objective (NA = 1, Leica). Homer-GCaMP3 was excited at 920 nm. A custom-built device was used as odor delivery system to supply odors with a constant flow rate of 1 ml/s to the fly's antennae for 2 s. Onset and duration of the odor stimulus were controlled using a custom-written LABVIEW program. Images were recorded at 10 Hz (Supplementary Fig. 11) or 33 Hz (Fig. 3). For in vivo imaging the following odors were used: benzaldehyde (diluted 1:500 in paraffin oil), 2, 3-butanedione (diluted 1:2000 in water), 4-methylcyclohexanol (1:500 in paraffin oil). Image processing and analysis were performed using Fiji software (1.52P). For correcting the potential slight movements in the *x–y* direction, recorded images were aligned using TurboReg plugin (Thevenaz, Ruttimann et al. 1998). Afterwards, regions of interest (ROIs) were manually defined in the calyx from two focal planes. The maximum five responsive microglomerular structures were selected as ROIs per animal. For signal quantification, the average pixel intensity of ten frames before stimulus onset was determined as $F_0$. $\Delta F$ is the difference between fluorescence and $F_0$, and resulting values were divided by $F_0$ and displayed as percent. Time to peak was measured from 10% to 90% of the maximum amplitude (Max $\Delta F/F_0$). False color-coded images were produced accordingly by subtracting the average of three frames before stimulus onset from the average of three frames after stimulus onset and dividing the resulting image by the average of the before stimulus. For display, images are median filtered with a pixel-range of five.

**Statistical analysis**. Data were tested for normal distribution using the Shapiro–Wilk test. Differences among multiple groups were tested by, one-way ANOVA with Bonferroni post hoc test. Differences between the two groups were tested by two-sample *t* test or Mann–Whitney *U* test. For gSTED analysis, the nonparametric Kruskal–Wallis test with the Dunn's post hoc test corrected for four multiple comparisons was applied to test differences in mean Unc13A/B-to-sfCac distances between groups. Differences in the frequency distribution of Unc13A/B-to-sfCac distances were tested with the two samples Kolmogorov–Smirnov test. Asterisks are used to indicate statistical significance of the results (n.s., not significant ($p > 0.05$), *$p \leq 0.05$, **$p \leq 0.01$, ***$p \leq 0.001$). The number of independent experiments ($n$) are mentioned in the figure legends.

**Reporting summary**. Further information on research design is available in the Nature Research Reporting Summary linked to this article.

## Data availability

All data supporting the findings of this study are provided within the paper and its supplementary information. All additional information and *Drosophila* lines generated for this study will be made available upon reasonable request to the corresponding author. Source data are provided with this paper.

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

## Acknowledgements
We thank the Bloomington Stock Center (BDSC) and Vienna Drosophila Resource Center (VDRC), for fly lines; and the Developmental Studies Hybridoma Bank (DSHB) for antibodies. We also thank the Core Facility BioSupraMol and SupraFAB of the Freie Universität Berlin for the use and support of the Leica TCS SP8 confocal microscope and Abberior Instruments STED Expert Line systems, respectively. We thank Dr. Alexander Walter and Dr. Niclas Gimber for providing the Macros and the Python script for STED analysis, respectively. We thank Carry Kuehnapfel, Anastasia Stawrakakis and Dr. Shubham Dipt for excellent technical assistance. This work was supported by grants (Forschergruppe 2705 Mushroom body; SFB1315, A08) from the Deutsche Forschungsgemeinschaft (DFG; German Research Foundation).

## Author contributions
A.P. and S.J.S. designed the project. A.P. performed *Drosophila* genetics, behavioral experiments, confocal imaging, and data analysis. A.P. and M.S. performed live imaging and A.P. and S.M.H. analyzed data. A.P. and T.M. performed the Airyscan microscopy and A.P. and M.M. analyzed data. M.M. and A.P. performed the STED imaging experiments and analyzed data. T.M., A.P., U.P., and A.F. generated transgenic lines. S.J.S. and A.P. wrote the manuscript with input from all the authors.

## Funding

## Competing interests
The authors declare no competing interests.
