## [Peer Review File · Nature Communications]

Reviewers' Comments:

Reviewer #1:

Remarks to the Author:

In this study, Pooryasin et al. investigated how the two Unc-13 isoforms are involved in the regulation of olfactory behavior in *Drosophila*. Unc-13 proteins are essential regulators of synaptic transmission, and play essential roles in synaptic vesicle docking, priming, and fusion. Synaptic transmission is completely arrested in mutant lacking Unc-13 proteins in almost all species such as mouse, fly, and the nematode *C. elegans*, demonstrating its conserved functionality. Unc-13 family proteins contain multiple functional domains and act as synaptic hubs interacting with several synaptic proteins. Moreover, Unc-13s are essential substrates in several cellular pathways and regulate key aspects of synaptic function, most notably short-term synaptic plasticity. Unc-13 family proteins consists of distinct isoforms, in which the isoform containing an N-terminal C2A domain (i.e., Munc13-1 in mouse, UNC-13L in worm, UNC-13A in fly) appears to play a dominant role in synaptic transmission. However, other isoforms are either co-expressed with the C2A-containing Unc-13s in the same neuron type or expressed in distinct neuron types. Those isoforms such as Munc13-2 in mouse, UNC-13S in worm, are also playing essential roles in synaptic transmission, but the SV release mediated by them appear to exhibit differential synaptic properties compared to the C2A-containing Unc-13s. Since being discovered, the functions of Unc-13 proteins have been extensively described in many model organisms, although the detailed mechanisms are still unclear, especially how Unc-13s are involved in behavioural regulation. The fruit fly is an ideal model to investigate this because of the ease of genetic manipulation and behavioural diversity. In this study, the authors revealed previously unknown functions of the two Unc-13 proteins, Unc-13A and Unc-13B. Knockdown of each individual Unc-13 protein produced differential effects in olfactory responses. The story is innovative by connecting different forms of short-term plasticity mediated by Unc-13 proteins to behavioural output, and provide interesting insights into the function of the nervous system.

Specific points:

1. Both Unc-13A and Unc-13B co-exist in ePNs and display differential distance to Ca²⁺ channels. The results are consistent with previous observations in other synapse types. How are their locations determined? Is the Unc-13 localization related to the protein domain structure?
2. It appears that Unc-13A and Unc-13B mediate synaptic vesicle fusion from distinct synaptic region (proximal and distal). Are the molecular organization at distinct areas (i.e., <70nm and >100nm) different? Are these two Unc-13 isoforms expressed in similar levels?
3. The readers may want to know how effective the RNAi used in this study. It will be good to show quantifications of Unc-13 protein levels after the RNAi. Does knockdown both Unc-13A and Unc-13B cause stronger decrease in transmission in Figure 2b?
4. The Munc13 isoforms exhibit differential functions in distinct synapse types. For example, Munc13-1 but not Munc13-2 is playing a dominant role in excitatory hippocampal neurons, and knockout of Munc13-1 almost abolishes EPSCs (Augustin et al., 1999; Varoqueaux et al., 2002). However, Munc13-1 and Munc13-2 are functionally redundant in inhibitory hippocampal neurons, and knockout of Munc13-1 or Munc13-2 does not produce a change in IPSCs. Does Unc-13A and Unc-13B also function redundantly in the iPN in fly? In figure 4, the author tested how knockdown of Unc-13B in iPN affect olfactory response. However, there is no evidence showing that Unc-13B RNAi in iPN cause decrease in synaptic transmission. If Unc-13A and Unc-13B do function redundantly in iPN synapses, it is not surprising to see that the aversive and appetitive responses are unaltered. How about knockdown Unc-13A in the iPN?
5. Since knockdown of Unc-13A or Unc-13B decreases transmission in similar extent (Figure 2b), it is likely that the differential effects of Unc-13A and Unc-13B RNAi on appetitive response arise from distinct patterns of transmission output (i.e., phasic and tonic), although the evidence is

lacking in PN output synapses. Does overexpression of Unc-13A (or Unc-13B) cause any phenotype in aversive and appetitive responses? Is there a possibility that the Unc-13A is increased by the knockdown of Unc-13B, thereby resulting in increased appetitive response?

Reviewer #2:

Remarks to the Author:

In the manuscript, by Dr. Pooryasin from the laboratory of Dr. Stephan J. Sigrist, the authors address an interesting problem of the organization of the unc13 family of proteins. The molecules the authors' study (UNC-13A and UNC-13B) are required for synaptic release but regulate different aspects of the process. The authors use a combination of super-resolution imaging and behavioral analysis of knockdowns in an attempt to link the nanoscale organization of these molecules to specific behaviors in the drosophila. This is an ambitious effort, and the authors make good progress toward their goal, however, most of their findings are correlative in nature, lessening the impact of the study.

The paper is broken into two sections that are not well integrated. In the first half of the paper, the authors use SIM and STED super-resolution imaging to examine the nanoscale organization of UNC-13A and UNC-13B. These studies are an interesting extension of the authors' previous work and demonstrate that these proteins are differentially organized in the fly brain. The second half of the paper examines the behavioral impact of knocking down these proteins in the cell population examined using super-resolution imaging. The behavioral impact of knockdown is interesting because there are different effects of knocking down UNC-13A vs. UNC-13B that may be related to their functions at the synapse. The manuscript would be significantly improved if the authors could determine whether the changes in behavior were at all related to nanoscale differences in presynaptic organization. This would seem to be a straightforward experiment. The authors also use dominant negative forms of UNC-13 proteins. Do these molecules cause similar changes to the nanoscale organization as the knockdown? Again, these are straightforward experiments that would speak to the issue of whether the effects of the knockdown relate to the nanoscale changes in synaptic structure. Simply showing changes at the confocal scale is not adequate.

The calcium imaging experiments are done at very low temporal resolution. The authors should conduct higher temporal resolution experiments to validate their claim that these slow imaging can accurately capture the events they claim to be studying. This is a significant concern as they may be missing key events that distinguish the function of UNC-13A from UNC-13B.

In a number of places, the paper uses strangely informal or unsupported language. For instance, the authors make seven statements using the word "obviously", including in the abstract. Obvious to whom? Does this mean there are statistically significant effects? Obviously is not a scientific term and must be removed. In most cases in this manuscript, the sentences would be unchanged in meaning if this word were removed, but if the authors are using the word to describe some statistical aspect of the findings, they must spell out what this is – for instance, do they mean an effect that is $p < 0.001$?

The last section of the paper titled "Unc13B-mediated transmission operates antagonistically in ePNs and iPNs" is very poorly put together. The writing is difficult to understand, overly informal, and not scientific, and the experimental design and results are not clearly explained. This section stands out from the rest of the paper and needs to be completely rewritten.

Reviewer #3:

Remarks to the Author:

The manuscript by Pooryasin et al. examines the distribution, physiological function, and behavioral significance of Unc13A and Unc13B in the terminals of ePNs of the *Drosophila* olfactory. The authors find that although knockdown of Unc13A or Unc13B induces similar levels of reduction in synaptic transmission, it causes dramatically different behavioral consequences in response to odor stimulation. Since the two proteins exhibit distinct coupling distances to voltage-gated calcium channels, which might contribute to different forms of short-term plasticity (STP), the authors conclude that the altered behavior is likely due to changes in the release properties of ePN terminals and thus the decoding of olfactory information. The paper is technically an outstanding study, combining clean genetics, super-resolution microscopy, functional imaging, and behavioral tests. It also provides evidence supporting the idea that fast dynamic adaptations in vesicle release regulate brain computation and behaviors. While the observations are fascinating, a few points will need to be addressed before the study is convincing.

- 1) The authors did an excellent job characterizing the nanoscale distribution of Unc13A and Unc13B in synaptic terminals and the behavioral consequences after knocking them down. Nevertheless, functional connections between the two observations are missing. In particular, the authors claim that the altered release pattern (or STP) might be the reason for the behavioral phenotypes, but they failed to provide any direct evidence. While performing a perfect PPR test using electrophysiology/calcium imaging might be challenging, altered calcium kinetics under train stimulation should be expected if STP has changed.
- 2) It is intuitive to assume that the distance between calcium source and calcium sensors (synaptotagmins) plays a significant role in determining the pattern/kinetics of vesicle release. However, it is not very clear how the coupling distance between calcium channels and Unc13s, proteins that mediate priming, contributes to release properties. The authors might need to provide more explanations on that.
- 3) It is hard to reconcile how fundamentally distinct behaviors are originated, given that the influences of removing either Unc13s on synaptic transmission are similar (fig. 2b and 4b). Is it possible that synapses other than the ones examined are actually underlying the tested behaviors?
- 4) What is the phenotype in synaptic transmission when both Unc13s are removed?
- 5) Does the nanoscale localization of the remaining Unc13 change after the other isoform is removed?
- 6) One would expect to see some quantifications of the Unc13 colocalization in fig. 1.

Response to reviewers:

Reviewer #1

1. Both Unc-13A and Unc-13B co-exist in ePNs and display differential distance to Ca²⁺ channels. The results are consistent with previous observations in other synapse types. How are their locations determined? Is the Unc-13 localization related to the protein domain structure?

We appreciate the opportunity to elucidate this point further and recognized that we should have given more space to this point in our original submission. Indeed, we previously showed that Unc13A is targeted into active zone central slots via interactions to scaffold protein BRP, whereas Unc13B is recruited into more active zone peripheral slots via the Syd-1 scaffold protein^{1,2}. Subsequent analysis² showed that these “targeting and positioning rules” apply across the *Drosophila* brain.

We do now write in the manuscript (page 4, line 49-53):

“By combining biochemical/genetic analysis with STED super-resolution microscopy, we showed that physical interactions with specific active zone scaffold proteins mediate these distinct coupling modes, with ELKS family Bruchpilot (BRP) targeting Unc13A but Syd-1 Unc13B.”

And (page 14, line 207-211):

“Our previous work² had shown that the AZ amounts of Unc13A and B do not scale with their sheer expression levels but are rather defined via the accessibility of their cognate AZ scaffold protein binding partners, BRP and Syd-1, respectively. Consistently, overexpression of *unc13A*-GFP or *unc13B*-GFP in ePNs did not alter olfactory responses (Supplementary Fig.7).”

2. It appears that Unc-13A and Unc-13B mediate synaptic vesicle fusion from distinct synaptic region (proximal and distal).

Apart from the STED analysis data¹⁻³ our previous biophysical and modelling analysis at NMJ synapses made a plausible case for the physical distance to indeed sufficiently explain the differences in short term plasticity and release probability between the Unc13A and B isoform.

Are the molecular organization at distinct areas (i.e., <70nm and >100nm) different?

As mentioned above, distinct largely “orthogonal” active zone scaffold components create binding specific slots for these isoforms. BRP localizes more towards the active zone center, and positions Unc13A, whereas Syd-1 localizes towards the active zone periphery and is responsible for Unc13B positioning. Notably, both these active zone scaffold proteins operate largely orthogonal means they in their localization and positioning do not depend on the respective other scaffold protein or the presence of their cognate Unc13 isoform.

Are these two Unc-13 isoforms expressed in similar levels?

We thank the reviewer for having brought up this important point which motivated us to perform a long-planned experiment. We here estimated the relative levels of Unc13A versus Unc13B in calyx and lateral horn by making use of a “self-made” antibody (“anti- Unc13C-term”) directed against the C-terminal “business end” common to both the Unc13A and B isoform. We used a pan-neuronal knockdown (via Appl-Gal4) of either isoform (Unc13A down to 0.19 ± 0.09 and Unc13B down to 0.04 ± 0.004 relative units) to calculate the level of corresponding

other isoform by measuring the remaining Unc13C-term signal. Our analysis shows that Unc13A and Unc13B levels are overall rather comparable, with Unc13B being somewhat more abundant than Unc13A in both calyx and lateral horn (Supplementary Fig. 3).

Consistently, also our new Ca²⁺ imaging data (Fig. 3) support the notion that per se both isoforms contribute to a similar quantitative extent to overall transmission of olfactory information, though with different timing characteristics.

These findings collectively also eliminate the possibility that the epistatic role of Unc13A over Unc13B in odor recognition merely could be a reflection of a per se higher contribution of the Unc13A isoform to release.

3. The readers may want to know how effective the RNAi used in this study. It will be good to show quantifications of Unc-13 protein levels after the RNAi.

Again, a very valuable point. To assess the efficiency of knockdown (KD), Unc13A and Unc13B intensities were now quantified as extent of protein knockdown in calyx (Fig. 2b) and lateral horn (Fig 2c); (Unc13A down to 0.38 ± 0.07 and Unc13B down to 0.24 ± 0.02 relative units).

According to these data, knockdown is about 60-75% efficient. That the extent is not too drastic we think is favorable for our analysis, given that already we could identify distinct behavioral phenotypes, but at same time do not interfere with the synaptic set up too drastically.

Importantly, KD of either isoform did not affect the level of the respective other isoform. Moreover, simultaneous double KD reduced both isoforms exactly as efficiently as single knockdown (Fig. 2b, c).

Does knockdown both Unc-13A and Unc-13B cause stronger decrease in transmission in Figure 2b?

An important question. Indeed, our new double KD of *unc13A* and *unc13B* lead to a further significant reduction in the amplitude of postsynaptic Ca²⁺ response when compared to both single KDs of *unc13A* or *unc13B* (Fig. 3). Indeed, all analysis in this study as well as in the studies before is fully consistent with the two release components operating in a truly orthogonal fashion.

4. The Munc13 isoforms exhibit differential functions in distinct synapse types. For example, Munc13-1 but not Munc13-2 is playing a dominant role in excitatory hippocampal neurons, and knockout of Munc13-1 almost abolishes EPSCs (Augustin et al., 1999; Varoqueaux et al., 2002). However, Munc13-1 and Munc13-2 are functionally redundant in inhibitory hippocampal neurons, and knockout of Munc13-1 or Munc13-2 does not produce a change in IPSCs. Does Unc-13A and Unc-13B also function redundantly in the iPN in fly? In figure 4, the author tested how knockdown of Unc-13B in iPN affect olfactory response. However, there is no evidence showing that Unc-13B RNAi in iPN cause decrease in synaptic transmission. If Unc-13A and Unc-13B do function redundantly in iPN synapses, it is not surprising to see that the aversive and appetitive responses are unaltered. How about knockdown Unc-13A in the iPN?

We are indeed really interested in testing for functional redundancy between the Unc13 isoforms in inhibitory neurons. As correctly indicated by the reviewer, down regulation of *unc13A* or *unc13B* in iPNs did not result in a significant change of olfactory responses (Fig. 6a, b). However, *unc13B* KD in iPNs was able to suppress the behavioral effect of

simultaneous *unc13B* KD in ePNs (Fig. 6c). To test for functional redundancy between *unc13A* and *unc13B* in iPNs, we now for the revised version generated new transgenic animals in order to be able to specifically down regulate *unc13A* in iPNs (using UAS/Gal4 system) together with *unc13B* KD in ePNs (using the QUAS/Q system). Notably, knockdown of *unc13A* in iPNs did not suppress the behavioral consequence of *unc13B* knockdown in ePNs (Fig. 6e). Thus, in the inhibitory iPNs, Unc13B seemingly performs functions non-redundant with Unc13A. Clearly, future analysis will have to address this interesting question in other cell types and additional behavioral assays.

5. Since knockdown of Unc-13A or Unc-13B decreases transmission in similar extent (Figure 2b), it is likely that the differential effects of Unc-13A and Unc-13B RNAi on appetitive response arise from distinct patterns of transmission output (i.e., phasic and tonic), although the evidence is lacking in PN output synapses. Does overexpression of Unc-13A (or Unc-13B) cause any phenotype in aversive and appetitive responses?

We fully understand the concern of the reviewer. We now recognized that we did not really describe our point here. We now corrected this and write in the revised manuscript (page 14, line 207-211):

“Our previous work ^{1,2} had shown that the AZ amounts of Unc13A and B, also across neuron types, do not scale with the sheer expression levels of the proteins but are rather defined via the accessibility of their cognate AZ scaffold protein binding partners, BRP and Syd-1, respectively. Consistently, overexpression of *unc13A-GFP* or *unc13B-GFP* in ePNs did not alter olfactory responses (Supplementary Fig.7).”

Is there a possibility that the Unc-13A is increased by the knockdown of Unc-13B, thereby resulting in increased appetitive response?

This certainly is per se a very good point and potential concern. However, we now did an experiment which excluded this possibility. We quantified Unc13A and Unc13B levels after single KD of Unc13A and Unc13B in calyx (Fig. 2b) and lateral horn (Fig 2c). Importantly, knockdown of either isoform did not affect the level of the respective other isoform, again supporting the “orthogonal character” of the two release components.

Reviewer #2

In the manuscript, by Dr. Pooryasin from the laboratory of Dr. Stephan J. Sigrist, the authors address an interesting and problem of the organization of the unc13 family of proteins. The molecules the authors' study (UNC-13A and UNC-13B) are required for synaptic release but regulate different aspects of the process. The authors use a combination of super-resolution imaging and behavioral analysis of knockdowns in an attempt to link the nanoscale organization of these molecules to specific behaviors in the drosophila. This is an ambitious effort, and the authors make good progress toward their goal, however, most of their findings are correlative in nature, lessening the impact of the study.

The paper is broken into two sections that are not well integrated. In the first half of the paper, the authors use SIM and STED super-resolution imaging to examine the nanoscale organization of UNC-13A and UNC-13B. These studies are an interesting extension of the authors' previous work and demonstrate that these proteins are differentially organized in the

fly brain. The second half of the paper examines the behavioral impact of knocking down these proteins in the cell population examined using super-resolution imaging. The behavioral impact of knockdown is interesting because there are different effects of knocking down UNC-13A vs. UNC-13B that may be related to their functions at the synapse.

The manuscript would be significantly improved if the authors could determine whether the changes in behavior were at all related to nanoscale differences in presynaptic organization. This would seem to be a straightforward experiment.

We thank the reviewer for having brought up this important point, which stimulated an as we think relevant experiment, which now is part of the main manuscript (Fig.5). In our strategy to manipulate nanoscale organization we now used an Unc13A-C-terminal fragment, whose effects we had studied in detail at NMJ synapses rendering themselves for biophysical analysis⁴. Here, this C-terminal fragment generated excessive release sites at atypical locations, means it interfered with proper nanoscale organization. This in turn resulted in severe changes of the release time course⁴. Conclusion here was that specific AZ localization via the N terminus prevents ectopic, temporally imprecise release.

In our revised version, we now expressed Unc13A-C-terminal fragment (*unc13A-C-term-GFP*) in ePNs (Fig. 5). As expected from our NMJ work, we observed that the Unc13A-C-terminal fragment efficiently reached the synaptic terminals, but instead of strongly enriching at the active zones (labeled with staining against endogenous full-length Unc13A) it formed ectopic clusters and smears. Most importantly, the Unc13A-C-terminal fragment expression provoked a very severe olfactory impairment (Fig. 5). These data suggest that indeed the nano-organization of Unc13 isoforms is indispensable for temporal fidelity and the release from Unc13A-C-term-GFP interfere with the precise Unc13A-dependent release.

The authors also use dominant negative forms of UNC-13 proteins. Do these molecules cause similar changes to the nanoscale organization as the knockdown? Again, these are straightforward experiments that would speak to the issue of whether the effects of the knockdown relate to the nanoscale changes in synaptic structure. Simply showing changes at the confocal scale is not adequate.

Previously, we showed that the *unc13A-N-term-GFP* fragment specifically interferes with Unc13A dependent release in a dominant negative fashion at larval neuromuscular synapses (NMJs)⁴. The N-term-fragment here interfered with the AZ recruitment of endogenous Unc13A by competitive binding to the AZ scaffold. In accordance with the previous study at NMJs⁴, STED analysis revealed that the N-term-GFP fragment when expressed in projection neurons formed discrete clusters (clearly different from the C-term-fragment) at seemingly identical “nano-location” as the endogenous “full-length” Unc13A (58.4 ± 1.2 nm from the BRP ring center) (Supplementary Fig.8).

We now write in the revised results part (Page 15, line 211-221): “An additional experiment was based on the observation that the number of Unc13 binding slots in the AZ scaffold is limited. Here, we overexpressed *unc13A-N-term-GFP* fragment in the ePNs. This fragment we previously showed to specifically interfere with Unc13A-dependent release in a dominant negative fashion at larval neuromuscular synapses (NMJs)⁴. The N-term-GFP fragment we here found to interfere with the AZ recruitment of endogenous Unc13A by competitive binding to the AZ scaffold. In accordance with a previous study at NMJs⁴, STED analysis revealed that the N-term-GFP fragment expressed in projection neurons clustered at a similar “nano-

location” as the endogenous “full-length” Unc13A (58.4 ± 1.2 nm from the BRP ring center). Indeed, overexpression of the unc13A-N-termGFP in the ePNs provoked the same behavioral deficit as unc13A knockdown (Supplementary Fig. 8). “

The calcium imaging experiments are done at very low temporal resolution. The authors should conduct higher temporal resolution experiments to validate their claim that these slow imaging can accurately capture the events they claim to be studying. This is a significant concern as they may be missing key events that distinguish the function of UNC-13A from UNC-13B.

We are very thankful for this valuable suggestion, which was shared by all reviewers and led to an important result (Fig.3).

We here as suggested increased the temporal resolution of intravital Ca^{2+} imaging from 10 Hz to 33 Hz when measuring the KC GCaMP3 signals after odor stimulation in *unc13A* or *unc13B* knockdown animals. When analyzing the kinetics of signal increase of odor presentation, we across different odors observed significant longer latencies to peak Ca^{2+} signal in *unc13A* KD and double knockdown animals, while *unc13B* KD did not influence the timing of signal increase compared to controls (Fig. 3)

We now write in the revised version: “Therefore, in order to characterize the timing of the Unc13A and Unc13B release components at PN output synapses, we analyzed the kinetics of the Ca^{2+} signals. As expected, both Unc13A and Unc13B to a similar degree contributed to the overall signal transfer (Max $\Delta F/F_0$ (%), Fig. 3b, g, l). Notably, analyzing the kinetics of odor onset across several odors, we observed significantly longer latencies to peak in *unc13A* knockdown and double knockdown animals, while *unc13B* knockdown did not influence the timing of signal onset compared to controls (Fig. 3d,e, i, j, n, o), These data provide direct evidence that also at ePN output synapses Unc13A specifically promotes a fast, transient component of SV release but Unc13B a more tonic release component. Thus, consistent with the specific AZ “nano-architectures” of Unc13A and B being highly similar across synapse types ^{2,5} (Fig. 1g-i), this specific filtering characteristics can be assumed to be present throughout the *Drosophila* brain. “

In a number of places, the paper uses strangely informal or unsupported language. For instance, the authors make seven statements using the word ‘obviously’; including in the abstract. Obvious to whom? Does this mean there are statistically significant effects? Obviously is not a scientific term and must be removed. In most cases in this manuscript, the sentences would be unchanged in meaning if this word were removed, but if the authors are using the word to describe some statistical aspect of the findings, they must spell out what this is; for instance, do they mean an effect that is $p < 0.001$? The last section of the paper titled ‘Unc13B-mediated transmission operates antagonistically in ePNs and iPNs’; is very poorly put together. The writing is difficult to understand, overly informal, and not scientific, and the experimental design and results are not clearly explained. This section stands out from the rest of the paper and needs to be completely rewritten.

We thank the reviewer for bringing up these points to our attention. We rephrased our statements and deleted the word obviously from the manuscript. In addition, we added more experiments to the ePNs and iPNs interaction section and add more explanation for better conveying our statements.

Reviewer #3

The manuscript by Pooryasin et al. examines the distribution, physiological function, and behavioral significance of Unc13A and Unc13B in the terminals of ePNs of the Drosophila olfactory. The authors find that although knockdown of Unc13A or Unc13B induces similar levels of reduction in synaptic transmission, it causes dramatically different behavioral consequences in response to odor stimulation. Since the two proteins exhibit distinct coupling distances to voltage-gated calcium channels, which might contribute to different forms of short-term plasticity (STP), the authors conclude that the altered behavior is likely due to changes in the release properties of ePN terminals and thus the decoding of olfactory information. The paper is technically an outstanding study, combining clean genetics, super-resolution microscopy, functional imaging, and behavioral tests. It also provides evidence supporting the idea that fast dynamic adaptations in vesicle release regulate brain computation and behaviors. While the observations are fascinating, a few points will need to be addressed before the study is convincing.

1) The authors did an excellent job characterizing the nanoscale distribution of Unc13A and Unc13B in synaptic terminals and the behavioral consequences after knocking them down. Nevertheless, functional connections between the two observations are missing. In particular, the authors claim that the altered release pattern (or STP) might be the reason for the behavioral phenotypes, but they failed to provide any direct evidence. While performing a perfect PPR test using electrophysiology/calcium imaging might be challenging, altered calcium kinetics under train stimulation should be expected if STP has changed.

We are very thankful for this valuable suggestion, which was shared by all reviewers and led to an important result.

We now increased the temporal resolution of intravital Ca^{2+} imaging from 10 Hz to 33 Hz when measuring the KC GCaMP3 signals after odor stimulation in *unc13A* or *unc13B* knockdown animals. When analyzing the kinetics of signal increase of odor presentation, we across different odors observed significant longer latencies to peak in the Ca^{2+} signals of *unc13A* KD and double knockdown animals, while *unc13B* KD did not influence the timing of signal increase compared to controls (Fig. 3)

We now write in the revised version: "Therefore, in order to characterize the timing of the Unc13A and Unc13B release components at PN output synapses, we analyzed the kinetics of the Ca^{2+} signals. As expected, both Unc13A and Unc13B to a similar degree contributed to the overall signal transfer (Max $\Delta F/F_0$ (%), Fig. 3b, g, l). Notably, analyzing the kinetics of odor onset across several odors, we observed significantly longer latencies to peak in *unc13A* knockdown and double knockdown animals, while *unc13B* knockdown did not influence the timing of signal onset compared to controls (Fig. 3d,e, i, j, n, o). These data provide direct evidence that also at ePN output synapses Unc13A specifically promotes a fast, transient component of SV release but Unc13B a more tonic release component. Thus, consistent with the specific AZ "nano-architectures" of Unc13A and B being highly similar across synapse types^{2,5} (Fig. 1g-i), this specific filtering characteristics can be assumed to be present throughout the *Drosophila* brain. "

2) It is intuitive to assume that the distance between calcium source and calcium sensors (synaptotagmins) plays a significant role in determining the pattern/kinetics of vesicle release. However, it is not very clear how the coupling distance between calcium channels and Unc13s,

proteins that mediate priming, contributes to release properties. The authors might need to provide more explanations on that.

We fully appreciate the point of the reviewer and now write in the introduction of the revised manuscript (Page 3, line: 30-44): “The molecular basis of STP is subject of intense investigation. Importantly, evolutionarily conserved AZ scaffolding proteins can determine the coupling distance between SV fusion sites and voltage-gated Ca^{2+} channels (VGCC) and, thereby, shape STP ⁶. Indeed, effective coupling distances vary across mammalian brain synapses, resulting in major functional differences ^{7,8}. Here, biophysical and electrophysiological analyses suggest that both release probability and STP depend greatly on the nanometer scale distance between synaptic vesicles (SVs) with their Ca^{2+} sensor Synaptotagmin, and voltage-operated Ca^{2+} channels (VGCCs) ⁷⁻¹². This is explained by the sharp spatiotemporal profile of action potential-induced Ca^{2+} transients and by the fact that SV fusion is activated by the cooperative binding of several (3-5) Ca^{2+} ions, resulting in a strong distance relationship for release probability: SVs positioned closer to the Ca^{2+} source have much higher release probabilities than distant ones. However, upon stimulation with higher action potential frequencies, synapses with high release probability tend to depress as the resupply of SVs becomes limiting, resulting in fast “phasic” release profile. In contrast, synapses with bigger coupling distance display a slower but sustained “tonic” release profile. Essential release factors of the m(Unc13) protein family thereby seem to define SV release sites and position SVs relative to the VGCCs ¹³.”

3) It is hard to reconcile how fundamentally distinct behaviors are originated, given that the influences of removing either Unc13s on synaptic transmission are similar (fig. 2b and 4b). Is it possible that synapses other than the ones examined are actually underlying the tested behaviors?

The reviewer is completely right that based on our old Ca^{2+} imaging data (Fig. 2b and 4b) with limited time resolution, no major difference between the effects of Unc13A or Unc13B KD was observable. As already mentioned above, we now increased the temporal resolution of intravital Ca^{2+} imaging from 10 Hz to 33 Hz when measuring the KC GCaMP3 signals after odor stimulation in *unc13A* or *unc13B* knockdown animals. When analyzing the kinetics of signal increase of odor presentation, we across different odors observed significant longer latencies to peak in the Ca^{2+} signals of *unc13A* KD and double knockdown animals, while *unc13B* KD did not influence the timing of signal increase compared to controls (Fig. 3). The reviewer is certainly right that we cannot fully exclude that other synaptic subpopulations might contribute to the effects we describe. However, the innate smell is processed at least very dominantly via lateral horn active zones. It in our eyes appears very likely that distinct release timing and STP behavior are indeed responsible for their distinct behavioral role.

In a similar context, we should emphasize that GH146-Gal4 is not absolutely exclusive for ePNs but also drives expression in the GABAergic anterior paired lateral (APL) neuron, with a single cell per hemisphere. To exclude a possible contribution of APL in the GH146-Gal4-Unc13A/B KD smell phenotypes, we down regulated *unc13A* or *unc13B* exclusively in APL using VT43924-Gal4. However, we did not observe any significant change in olfactory response in these animals in comparison to the control groups (Supplementary Fig. 6), excluding a major role of this neuron type in our analysis.

4) What is the phenotype in synaptic transmission when both Unc13s are removed?

Double KD of *unc13A* and *unc13B* lead to a further reduction in the amplitude of postsynaptic Ca^{2+} response when compared to both single KD of *unc13A* or *unc13B* at PN::KC synapses in the calyx (Fig. 3). Thus, obviously both components fulfill non-redundant functions at this synapse.

5) Does the nanoscale localization of the remaining Unc13 change after the other isoform is removed?

We shared the concern of the reviewer, and performed an additional STED experiment. Now in the revised manuscript, in *unc13A* KD and *unc13B* KD animals we measured the localization of the respective other isoform relative to the BRP ring centers (identifying the center of the AZ and center of Ca^{2+} channel) in calyx ePN boutons (Fig. 2d, e). STED analysis revealed that the nanoscopic localizations are not affected by the reduced physical presence of the respective other isoform (Fig. 2f). Thus, consistent with their distinct molecular targeting logic², both isoforms and release components in an orthogonal fashion.

6) One would expect to see some quantifications of the Unc13 colocalization in fig. 1.

As the reviewer suggested we calculated Mander's coefficient values between the BRP-short^{GFP} label and Unc13A or Unc13B (Supplementary Fig. 1). The Mander's coefficient values were very high between the BRP-short^{GFP} label and both Unc13A (in calyx: 0.8 ± 0.02 , in lateral horn: 0.76 ± 0.01) and Unc13B (in calyx: 0.88 ± 0.02 , in lateral horn: 0.77 ± 0.04) in ePNs (Supplementary Fig. 1). The Mander's coefficient values in iPNs showed a somewhat lower level of co-occurrence between BRP-short^{GFP} and Unc13B (0.41 ± 0.01) than BRP-short^{GFP} and Unc13A (0.66 ± 0.02) (Supplementary Fig. 1b).

References

- 1 Bohme, M. A. *et al.* Active zone scaffolds differentially accumulate Unc13 isoforms to tune Ca^{2+} channel-vesicle coupling. *Nature neuroscience*, doi:10.1038/nn.4364 (2016).
- 2 Fulterer, A. *et al.* Active Zone Scaffold Protein Ratios Tune Functional Diversity across Brain Synapses. *Cell reports* **23**, 1259-1274, doi:10.1016/j.celrep.2018.03.126 (2018).
- 3 Woitkuhn, J. *et al.* The Unc13A isoform is important for phasic release and olfactory memory formation at mushroom body synapses. *Journal of neurogenetics*, 1-9, doi:10.1080/01677063.2019.1710146 (2020).
- 4 Reddy-Alla, S. *et al.* Stable Positioning of Unc13 Restricts Synaptic Vesicle Fusion to Defined Release Sites to Promote Synchronous Neurotransmission. *Neuron* **95**, 1350-1364 e1312, doi:10.1016/j.neuron.2017.08.016 (2017).
- 5 Bohme, M. A. *et al.* Active zone scaffolds differentially accumulate Unc13 isoforms to tune Ca^{2+} channel-vesicle coupling. *Nature neuroscience* **19**, 1311-1320, doi:10.1038/nn.4364 (2016).
- 6 Bohme, M. A., Grasskamp, A. T. & Walter, A. M. Regulation of synaptic release-site Ca^{2+} channel coupling as a mechanism to control release probability and short-term plasticity. *FEBS letters* **592**, 3516-3531, doi:10.1002/1873-3468.13188 (2018).
- 7 Eggermann, E., Bucurenciu, I., Goswami, S. P. & Jonas, P. Nanodomain coupling between Ca^{2+} channels and sensors of exocytosis at fast mammalian synapses. *Nature reviews. Neuroscience* **13**, 7-21, doi:10.1038/nrn3125 (2011).

- 8 Rebola, N. *et al.* Distinct Nanoscale Calcium Channel and Synaptic Vesicle Topographies Contribute to the Diversity of Synaptic Function. *Neuron* **104**, 693-710 e699, doi:10.1016/j.neuron.2019.08.014 (2019).
- 9 Stanley, E. F. The Nanophysiology of Fast Transmitter Release. *Trends in neurosciences* **39**, 183-197, doi:10.1016/j.tins.2016.01.005 (2016).
- 10 Vyleta, N. P. & Jonas, P. Loose coupling between Ca²⁺ channels and release sensors at a plastic hippocampal synapse. *Science* **343**, 665-670, doi:10.1126/science.1244811 (2014).
- 11 Wadel, K., Neher, E. & Sakaba, T. The coupling between synaptic vesicles and Ca²⁺ channels determines fast neurotransmitter release. *Neuron* **53**, 563-575, doi:10.1016/j.neuron.2007.01.021 (2007).
- 12 Wang, S. S. H. *et al.* Fusion Competent Synaptic Vesicles Persist upon Active Zone Disruption and Loss of Vesicle Docking. *Neuron* **91**, 777-791, doi:10.1016/j.neuron.2016.07.005 (2016).
- 13 Reddy-Alla, S. *et al.* Stable Positioning of Unc13 Restricts Synaptic Vesicle Fusion to Defined Release Sites to Promote Synchronous Neurotransmission. *Neuron*, doi:10.1016/j.neuron.2017.08.016 (2017).

Reviewers' Comments:

Reviewer #1:

Remarks to the Author:

The authors have addressed all my questions and concerns. I recommend publication of this manuscript in Nature Comms.

Reviewer #2:

Remarks to the Author:

The authors have carefully address my concerns and the concerns of the other reviewers. In particular the two parts of the paper are much better integrated and read better. The manuscript is important and will be of interest to many in the field of neuroscience and synaptic biology.

Reviewer #3:

Remarks to the Author:

The authors have addressed all my concerns and I support its publication in Nature Communications

Reviewer #4:

Remarks to the Author:

In the manuscript, Dr. Pooryasin et al. reported an interesting finding that the two vesicle priming factors, Unc-13A and 13B, could transmit distinct olfactory information. The authors obtained three sets of results: 1) ultrastructural difference between the two Unc-13s in PN terminals (High quality, but incremental); 2) synaptic transmission with calcium imaging approach (Poor quality and inadequate); 3) behavioral observation that attraction is enhanced in Unc-13B knockdown flies (Novel and interesting). However, these three sets of data do not provide a coherent picture to mechanistically understand the roles of Unc-13s in behavior filtering. The manuscript could be further improved by addressing the following concerns. Otherwise, the manuscript should not be published in this journal.

Major concerns

- 1) To quantify calcium dynamics, a full dose of odor stimulation is needed since dynamics is related to amplitudes in a non-linear manner. Without measurements from a defined KC population, a simple average from random KC measurements may mask the true mechanism if KC show different changes.
- 2) More importantly, it is the LHNs, but not KCs, that mediate innate behavior. Thus, functional examination of LHNs, instead of KCs, should be performed. Furthermore, the slow calcium imaging data could be replaced by the fast voltage imaging data since the tools such as Voltron are available. Also, LHN measurements could directly test the authors' claim that LHNs integrate tonic and phasic responses.

Minor concerns

- 1) Quantification in Figure S3 needs to consider the low efficiency of knockdown.
- 2) Are Unc-13s also expressed in other neurons, such as local neurons in antennal lobe, lateral horn neurons, etc? How would these neurons contribute to behavior filtering by Unc-13s?
- 3) Are Unc-13s differentially expressed in different PNs (one extreme case: 13B is expressed only in PNs that drive avoidance)?

Reviewer #4 (Remarks to the Author):

In the manuscript, Dr. Pooryasin et al. reported an interesting finding that the two vesicle priming factors, Unc-13A and 13B, could transmit distinct olfactory information. The authors obtained three sets of results: 1) ultrastructural difference between the two Unc-13s in PN terminals (High quality, but incremental); 2) synaptic transmission with calcium imaging approach (Poor quality and inadequate); 3) behavioral observation that attraction is enhanced in Unc-13B knockdown flies (Novel and interesting). However, these three sets of data do not provide a coherent picture to mechanistically understand the roles of Unc-13s in behavior filtering. The manuscript could be further improved by addressing the following concerns. Otherwise, the manuscript should not be published in this journal.

Response:

We are happy to learn that our behavioral analysis was judged “novel and interesting” by the reviewer. Concerning the comments regarding the imaging quality and adequacy, we feel that we did not manage to fully convey the purpose of our experiments as well as the protocols used in comprehensive manner, also due to space constrains. We thus appreciate the chance to here in more detail explain our experimental strategies.

Please find below our reply on the specific points of the reviewer. We here took the liberty to group these points regarding whether they concerned questions related our imaging protocols or the question related to the in vivo imaging of lateral horn neurons.

Major concerns of reviewer#4 regarding imaging protocols

1) “To quantify calcium dynamics, a full dose of odor stimulation is needed since dynamics is related to amplitudes in a non-linear manner.”

We fully understand that the reviewer wants to make sure that our results are robust against details of the experimental setup and appreciate the possibility to present an in-depth analysis of our data.

Principal aim of the Ca²⁺ imaging experiment **first** was to define the **relative contributions of Unc13A and Unc13B to SV release at PN output synapses with single bouton resolution**. For this purpose, we took advantage of the unique “microglomerular” organization in the mushroom body calyx, where discrete ePN presynaptic boutons are directly connected to the postsynaptic specializations of Kenyon cells (KCs) dendrites (“claws”). Therefore, we measured Ca²⁺ response at these KC postsynaptic specializations in animals with either *unc13A* or *unc13B* knockdown in the ePNs^{1,2}. Clearly, both individual knockdown scenarios significantly reduced postsynaptic KC Ca²⁺ signals, and reductions behaved additively, proving that indeed both *unc13* species individually contribute to synaptic release here (Fig. 3).

Second, we investigated the **release latencies** (time to peak) after *unc13A* or *unc13B* knockdown. Now in the revised version, we added more description about our live imaging analysis in the material and method section.

Our principal intention when selecting appropriate odor concentrations was to use **the same odor concentration conditions in performing in vivo imaging of synaptic transmission and behavioral experiments**. Also, as we have shown in the manuscript (Supplementary Fig. 6), increasing the odor concentration did restore the odor avoidance after both *unc13A* and *unc13B* knockdown (Supplementary Fig. 6). However, the reviewer is certainly right that per se the “dynamics (of Ca²⁺ signals) is related to amplitudes in a non-linear manner”.

Rebuttal Fig 1. **No correlation between release kinetics (latencies) and Ca^{2+} signal amplitudes.** The maximum amplitude ($\Delta\text{F}/\text{F}_0$) plotted against time to peak (latency) for each responsive microglomerulus for control (*GH146-Gal4 / +; mb247: homer-GCaMP3 / +*), *unc13B*-knockdown (*GH146 > unc13B-RNAi; mb247: homer-GCaMP3*) and *unc13A*-knockdown (*GH146 > unc13A-RNAi; mb247: homer-GCaMP3*) groups. The Pearson's correlation scores show no correlation between amplitude and the latency in any group. Odor stimulation: 4-methylcyclohexanol. n=24-42 microglomeruli.

The obvious question here for us is whether the release kinetic differences were indeed a function of Ca^{2+} signal amplitudes.

Thus, we now performed an in-depth analysis of our live imaging data, correlating Ca^{2+} signal amplitudes ($\Delta\text{F}/\text{F}_0$ values) with the time to peak values at individual microglomerulus level. As can be seen from Rebuttal Fig. 1, **release kinetics (latencies) and Ca^{2+} signal amplitudes ($\Delta\text{F}/\text{F}_0$)** were uncorrelated over nearly a 5-fold range of Ca^{2+} signal intensities. We thus conclude that in the physiologically relevant odor concentration range we chose, kinetics of the Ca^{2+} signals are obviously not systematically influenced by local odor driven signal intensities ($\Delta\text{F}/\text{F}_0$ amplitudes).

Concerning the distinct role of *Unc13A* and *Unc13B* for release kinetics (*Unc13A* phasic versus *Unc13B* promoting tonic release), we also would like to emphasize that **we previously demonstrated this distinct behavior at three other *Drosophila* synapse types:** glutamatergic NMJ synapses³, ORN:PN synapses⁴ and KC:MBON synapses⁵. Given this body of evidence, we think that the robust

timing difference we obtained after *Unc13A* or *Unc13B* knockdown indeed strongly indicates that this behavior is equally present across all PN output synapses. Importantly here, our STED analysis (Fig. 1) showed that the biophysical reason for this difference – different nanospacing of *Unc13* and *Unc13B* versus the AZ central Ca^{2+} channels – is equally present at all PN output synapses, regardless of whether in the calyx or the lateral horn (LH) (Fig. 1). While as described below a truly reliable analysis of release kinetics at ePN::LH synapses is, despite our substantial efforts, currently precluded, we think that it is **more than likely that the release kinetic differences between *Unc13A* and *Unc13B* are of generic nature means obviously present across synapse types.**

We would again like to emphasize that we do have clearly demonstrated release kinetics differences between *Unc13A* and *Unc13B* at one population of ePN putput synapses. Also given that the ePN synapses in LH should “see” the very same distribution of action potential frequencies as the calyx ePN synapses, we think that it is safe to conclude that these synapses will behave at least very similarly. This also given that ePN synapses in calyx and LH show essentially the same nanoarchitecture (Fig. 1).

2) Without measurements from a defined KC population, a simple average from random KC measurements may mask the true mechanism if KC show different changes.

Rebuttal Fig 2. **Signal intensities and peak latencies across different odor responsive microglomeruli are normally distributed.** Frequency histogram for maximum amplitude ($\Delta F/F_0$) (a) and time to peak (latency) (b) for control (*GH146-Gal4 / +; mb247: homer-GCaMP3 / +; unc13B-RNAi; mb247: homer-GCaMP3*) and *unc13A*-knockdown (*GH146 > unc13A-RNAi; mb247: homer-GCaMP3*) groups. Odor stimulation: 4-methylcyclohexanol. n=24-40 microglomeruli. Normality test: Shapiro-Wilk test.

We measured Ca^{2+} transients in the confined subset of microglomeruli being responsive to a given odor, and then averaged responses for each animal.

As we did not measure in a “random fashion” but measure from those KCs whose dendrites are responsive, we simply would not know how to further restrict the KCs sampled in a meaningful manner. In fact, focusing on responsive microglomeruli rather than

selecting them in a random manner appears as the most appropriate method in our eyes.

In addition, given that the odor-representations on KC level are not truly stereotyped in between individual animals, we are convinced that the strategy chosen here was appropriate. The reviewer, for a good reason, asked whether there would be **systematic differences in between different PNs and their release kinetics** (“latencies”). We thus re-analyzed our data on single microglomerulus level. Notably, time to peak data for individual microglomeruli were narrowly distributed and followed a normal distribution, indicating that there indeed are no different PN classes concerning distinct release timing properties being present in the calyx (Rebuttal Fig. 2).

We are thankful to the reviewer for having raised this concern, as it showed to us that we did not appropriately document our approach. To now in the revised manuscript ease access for the reader, we combined Rebuttal Figures 1 and 2 as a new figure (Supplementary Fig. 5) to the revised paper.

We now write in the manuscript:

(Line 187-193) No correlation between release kinetics (latencies) and Ca^{2+} signal amplitudes ($\text{Max } \Delta F/F_0$) for each responsive microglomerulus was observed. Thus, kinetics of the Ca^{2+} signals are evidently not systematically influenced by local odor driven signal intensities (Supplementary Fig. 5 a-c). In addition, in each experimental group, the release kinetics and Ca^{2+} signal amplitudes measured at single microglomerulus level were normally distributed (Supplementary Fig. 5 d-i). Our analysis thus suggests that there are no major disparities between different types of KCs here.

To further make sure that we representatively sample over different KC types, we performed a new experiment, measuring odor-induced Ca^{2+} responses in three different focal planes in the calyx to be able to cover a broad spectrum of microglomeruli (Rebuttal Fig. 3). As you can see from Rebuttal Fig. 3, Ca^{2+} signal kinetics showed no significant differences in between different spatial areas of the calyx given by the three different focal planes chosen. Thus, while the reviewer correctly asks whether there might be systematic differences between KC Ca^{2+} signals for different KCs, our new additional analysis strongly indicates that this indeed is not the case (Rebuttal Fig. 3).

Rebuttal Fig 3. **Odor-induced Ca^{2+} signal kinetics showed no significant differences between different spatial areas of the calyx.**

a Representative two-photon images of odor-induced fluorescence changes of GCaMP3 in mushroom body calyx (mb247: homer-GCaMP3) at the dendritic specializations of KCs ("microglomeruli") at three focal planes. **b** Comparison between odor responsive microglomeruli selected from three different focal planes does not show any difference in time to peak (ms). **c** Maximum

amplitude ($\Delta F/F_0$) plotted against time to peak (latency) for each odor responsive microglomerulus sampled from three focal planes. The Pearson's correlation scores show no correlation between amplitude and the latency. Box plots indicate means, medians, interquartile ranges, and SD ranges. $n = 36$ microglomeruli, 4 animals. odor stimulation: 4-methylcyclohexanol. P-values were calculated via one-way ANOVA with the Bonferroni multiple comparison post hoc test (n.s., not significant). The yellow circles indicate the odor responsive microglomeruli.

3) "Furthermore, the slow calcium imaging data could be replaced by the fast voltage imaging data since the tools such as Voltron are available."

Concerning sampling speed:

The reviewer rightly emphasizes that one aspect of choosing an imaging reporter and an appropriate sampling speed is that this setting has to be suitable to discriminate the relevant timing differences, here obviously the different timing in between the Unc13A and B release components.

We thus now in response to the reviewer's question analyzed the influence of sampling speed on the quantification of release kinetics in wild type animals, by comparing 33 Hz sampling speed (used in manuscript) to 61 Hz, asking whether quantification of release timing would change.

As can be seen from Rebuttal Fig. 4b-d, while signal noise still allowed imaging at 61 Hz, the measured timing of release (latency) was not affected when using 61 Hz imaging. Seeing this saturation here, we think that a further increase of sampling speed would not change our results, at least qualitatively. We are therefore convinced that the latencies measured here are determined by the true biological response time of the olfactory system needed for signal processing and synaptic transmission.

Rebuttal Fig. 4.

Comparison of odor-induced Ca^{2+} signal kinetics between different sampling speed. **a** Representative images of in vivo calcium imaging of homer-GCaMP3 using two-photon microscopy with different sampling speed in one animal (same focal plane). **b** Odor-induced fluorescence change ($\Delta F/F_0$ %) of GCaMP3 in animals expressing

mb247: homer-GCaMP3 imaged at 33 Hz and 61 Hz. **c, d** No difference in Max $\Delta F/F_0$ (%) and time to peak (ms) between 33 Hz and 61 Hz groups was observed. Box plots indicate means, medians, interquartile ranges, and SD ranges. n= 36 microglomeruli, 4 animals. odor stimulation: 4-methylcyclohexanol. P-values were calculated via two-sample t-test (n.s., not significant).

Concerning choice of sensor:

We fully agree with the reviewer that the right choice of imaging sensor is critical when recording synaptic signals. Here, to record synapse-activity driven Ca^{2+} responses specifically at postsynaptic specializations, we expressed a GCaMP3 fusion to postsynaptic protein Homer, (instead of using a cytosolic GCaMP3), previously designed⁶ by Dr. André Fiala, co-author of the study. As when expressed in KCs, Homer targets into the actual postsynaptic specialization directly opposite the presynaptic active zones (Supplementary Fig. 3). This sensor thus restricts the signal to Ca^{2+} signals (evoked by local depolarization and direct Ca^{2+} flux through acetylcholine receptors) very close to the presynaptic release sites.

We are afraid that a similar localized targeting of the FRET based membrane voltage sensors such as Voltron ⁷, necessarily being transmembrane proteins, has not been possible (despite efforts). Moreover, we would like to emphasize that $\Delta F/F_0$ values of transgenic voltage sensors such as Voltron (less than - 5 %-reported in *Drosophila*) ⁷ are still considerable below the ones of Ca²⁺ sensors such as GCAMPs (100% or more). Thus, despite the fact that Voltron can pick individual action potentials, we are convinced that the Homer-GCAMP fusion here is a fully appropriate choice and that the timing differences we detect are obviously mechanistically and biologically meaningful, also in the light of our previous work detecting this principal difference across a spectrum of synapses ^{4,8,9}.

In addition, for using Voltron, prior to the in vivo imaging it would take two hours of sample preparation including 1 hour incubation with the Janelia Fluor (JF) dye in the alive, open headed fly, dye washout and subsequently 1 hour saline incubation. This long procedure we are afraid would complicate the olfactory-based in vivo imaging ⁷.

Finally, the most important point here is that despite extensive improvement of Voltron regarding higher basal fluorescence, lower photobleaching, and better signal to noise ratio, **opsin-based sensors such as Voltron are not yet compatible with two-photon imaging** ^{7,10,11}. Compatibility with two-photon microscopy, however, is essential for monitoring neuron activity in compact tissues ¹² like the calyx here.

2) More importantly, it is the LHNs, but not KCs, that mediate innate behavior. Thus, functional examination of LHNs, instead of KCs, should be performed. Also, LHN measurements could directly test the authors' claim that LHNs integrate tonic and phasic responses.

We thank the reviewer for bringing up this point, as it gives us the chance to in detail describe our strategy as well as our extensive efforts in this regard, and why such experiments currently cannot be performed in the LH in any quality close to the data we retrieved in calyx.

Before describing our efforts below, we would like to emphasize that we did not claim to have deconstructed the computational processes, neither in the LH nor in the mushroom body, which finally do integrate distinct behavioral components downstream of the synaptic filtering executed by Unc13A and Unc13B dependent release. That such an analysis will have to be subject of future work we think was clear throughout the review process.

As formulated above already, we provided a body of evidence that by all likelihood also PN output synapses are no exception from Unc13A mediating a phasic and Unc13B a tonic release component. As correctly phrased by the reviewer, our analysis of the role of ePNs and iPNs indeed suggests the lateral horn as relevant for signal integration in the behavioral response we describe (Fig. 6). Still, we would also like to add that **mushroom body and lateral horn cannot be exclusively classified as relevant for learned and naïve responses, respectively** ¹³⁻¹⁶. Notably, we in our paper showed that output of KCs is indeed essential for aversive response in our behavioral paradigm (Supplementary Fig. 12). We sure agree with the reviewer, however, that an in depth understanding of signal integration in the LH, which unfortunately must be beyond the scope of a first paper describing a fundamental new phenomenon, will be a logical and necessary next step to be taken.

In this regard, it is important to note that in order to really **characterize the release properties of ePN output synapses by attenuating either Unc13A or Unc13B, via a postsynaptically localized sensor, we needed to perform Ca²⁺ imaging at defined synaptic terminals where beyond doubt ePNs directly connect to LH principal neurons**. In the experiments we performed, the stereotyped cytoarchitecture of the calycal microglomeruli, *bona fide* connecting ePNs with cognate KC postsynaptic specializations, is a very "fortunate context

and morphological landmark” which allows for such an approach and was thus extensively exploited in our study. To rephrase it, it is the stereotyped synaptic ultrastructure of the calycal microglomerulus organization which enables straightforward experiments here.

Unfortunately, however, the situation is fundamentally different in the LH and morphological landmarks reporting where **ePNs directly connected to LH principal neurons are simply absent here**. LHNs receive odor information from both excitatory projection neurons (ePNs) and inhibitory projection neurons (iPNs) with form an “intermingled architecture” in the LH region (Rebuttal Fig. 5a).

As we are sure the reviewer is aware of, the LH entails about 860 LH output neurons (LHONs) and 560 local neurons (LHLNs) ¹⁷⁻¹⁹. Therefore, in contrast to the calyx, lateral inhibition must be a dominant feature of signal computation of the the LH ¹⁷, and recent connectomic reconstructions of olfactory PNs revealed a highly complex connectivity diagram **with prominent LH:LH neurons connections**, also via axoaxonic connections ¹⁷ (Rebuttal Fig. 5b) .

Rebuttal Fig 5. **Projection neuron outputs and principal wiring diagram of the lateral horn region** a Confocal image representing projection of ePNs (green) and iPNs (red) neurons in LH. Scale bar: 10 μ m. b Wiring diagram for LH neuropil, in terms of input, local, output and feedback neurons ¹⁷.

In contrast to distinctive anatomical feature of ePN-KC synapses in the calyx with the well-defined microglomerular structure, ePN-LHN synapses unfortunately do not share the same characteristic stereotypies, and consequently the absence of such a landmark makes it much more complicated to reliably define postsynaptic partners of ePNs with unc13 isoform downregulation. To further complicate the issue, up to now there are **no generic LH neuron drivers** available.

In pilot experiments we had performed previously (Rebuttal Fig. 6), upon methylcyclohexanol (MCH) or benzaldehyde (BA) stimulations, different regions of LHN arborization covered by the GMR16C09-Gal4 driver indeed displayed Ca^{2+} responses. However, due to the reasons described above, detecting exact ePN bouton::LHN connections was impossible using this approach. In other words, we unfortunately using such an approach would simply not know whether we observed a direct ePN-triggered Ca^{2+} response or an indirect response involving iPN::LHN or LHN:LHN communication. The last scenario, however, would obviously undercut our interpretation concerning the knockdown results, as the knockdown obviously applies to only the ePNs but intentionally not the other neuron types which, however, would become involved in signal computation.

Rebuttal Fig 6. **Representative two-photon images of odor-induced fluorescence changes of GCaMP3 in LH in animal expressing GCaMP3 under control of GMR16C09.** White areas indicate LH region. Red areas indicate odor-induced Ca^{2+} response upon 4-methylcyclohexanol (MCH) stimulation. Yellow areas indicate odor-induced Ca^{2+} response upon Benzaldehyde (BA) stimulation. Scale bar: 10 μm .

Since our goal was to look at the release mechanism of ePN output neurons after manipulating the level of *unc13* isoforms in ePNs and as a read out measure the odor-induced Ca^{2+} in LHNs, we searched for Gal4 driver line covering substantial number of LHNs.

We intended to generate new transgenic lines, LH-enhancer::*homer-GCaMP*, to express *homer-GCaMP* at postsynaptic specializations of LH neurons. Such lines in combination with orthogonal expression of e.g. a RFP in the ePN terminals would have maybe allowed to at least at confocal level identify ePN::LHN contacts. Unfortunately, despite having screened many lines (some examples in Rebuttal Fig. 7), however, we could not find any appropriate line driving the expression in many LHNs. Instead, all Gal4 lines also were very “dirty” (means expressed in many other neuron types. Unfortunately, it often was impossible to distinguish between proper LHNs arborization and other type of neurons in the LH region.

Rebuttal Fig 7. **Expression pattern of several Gal4 driver lines for LHNs tested by us.** Images represent maximal projection fluorescence intensity across a confocal stack of brains, immunostained against GFP (green) and BRP^{NC82} (red). Scale bar: 50 μ m.

Notably, the available **LHN split Gal4 collection**¹⁸ lines display LHN specific expression. Unfortunately, however, each individual line here covers only very few LHNs (less than 10 from 1400 LHNs). We are afraid that investigating such a low number of neurons would bear the risk of not retrieving a representative result. More importantly, however, also when using split Gal4 lines, it equally would not necessarily have been possible to discriminate signals reflecting direct ePN:LHN synaptic transmission from signals involving iPN::LHN or LHN:LHN communication. **Despite these issues, we made an extensive effort to be able to downregulate *unc13* isoforms in ePNs and also label them by expressing RFP while at the same time orthogonally expressing the imaging sensor via Split-gal4 lines.**

To knockdown *unc13A/B* we tested these two approaches:

1. LexA/LexAop system 2. QF/QUAS system

LexA/LexAop approach

In this approach we planned to express mRFP and knocking down *unc13* isoforms in ePNs using LexA/LexAop system and at the same time express homer-GCaMP3 in LHNs using split-Gal4 lines. We thus established two new transgenic lines, LexAop:*unc13A*-RNAi and LexAop:*unc13B*-RNAi, to be able to downregulate these proteins using GH146-LexA. However, as GH146-LexA drives only weakly, we were not able to downregulate *unc13A* or *unc13B* sufficiently in ePNs using this approach (Rebuttal Fig. 8). We have evaluated the LexAop:*unc13A*-RNAi and LexAop:*unc13B*-RNAi lines using other types of LexA driver lines (e.g. mb247-LexA), and here downregulation was efficient. Since GH146-lexA however was not driving sufficiently, this approach unfortunately failed.

Rebuttal Fig 8. Confocal images from the calyx region in animals expressing LexAop: *unc13B-RNAi* under control of GH146-LexA, compared to heterozygous driver line, GH146-LexA (control), immunostained against Unc13B (green). Scale bars: 10 μ m.

QF/QUAS approach

In this approach, the strategy was to express the RNAi and QUAS-mtdTomato-3xHA²⁰ using GH146-QF and express homer-GCaMP3 in LHNs using split-Gal4 lines. Thus, we here generated a QUAS: *unc13A-RNAi* and a QUAS: *unc13B-RNAi* transgenic line. We tested the efficiency of these lines in immunostaining and behavior (Fig. 6d and Supplementary Fig. 13). Although these lines were suitable for behavior experiments, we could not use them for dual color live imaging. The GH146-QF line (w[1118]; PBac{Disc\RFP[DsRed2.3xP3]=GH146-QF) expresses dsRed under control of 3xP3 promoter (which cannot be excised out). This promoter drives a very strong expression in optical lobes but also displays very high “background” in the central brain as well. Therefore, it was not possible to use the low expression of mtdTomato-3xHA as a suitable landmark of ePN output terminals.

Thus, despite all these efforts, we were not able to perform live imaging in LHNs with quality and reasonable standards required for measuring the release kinetics of ePN output neurons. Notably, there is only a scarce number of publications which performed odor-induced live imaging at LHN level^{21,22}, and importantly no studies which would have directly measured synaptic signals or even kinetics of synaptic release in the LH. In the few previous papers performing olfactory live imaging in LH neurons, either Ca²⁺ responses in cell bodies²¹ or signals averaged over the whole LHN arborization were measured²². As we explained extensively that this type of analysis is unfortunately not suitable for our aim, however, as we need a biophysical measurement at the exact synaptic terminals where *unc13* levels were attenuated. We are thus afraid that a generic measurement of LHNs odor-driven activity signals in the LH necessarily entailing different types of direct and indirect connections would not be really fruitful. We sure would be happy to learn about options here which escaped our attention.

A direct presynaptic measurement of synaptic vesicle fusion at ePN output synapses in LH would per se be an alternative approach. In an interesting previous study²³, Parnas et al. showed that iPN inhibition imposes a high-pass filter on ePN synaptic output in LH by measuring synaptophluorin (sPH) fluorescence changes at ePN synaptic terminals present in the the LH region, evoked by electrical stimulation. However, the very low sPH signal levels reported here at ePN output synapses (less than 1% $\Delta F/F_0$)²³ makes also this approach not appropriate for measuring the kinetics of synaptic release, particularly given the very low signal to noise ratio.

We would like to thank the reviewer for this helpful comment that made us aware that in the manuscript we have not made this point about LH clear yet. In the improved version of the manuscript, now we add more explanation about difficulties of LHNs measurement for our purpose.

We now write in the manuscript:

(Line 160-165) As the lateral horn is considered to play a major role in generating innate responses to olfactory cues, measuring ePN release characteristics at lateral horn neurons (LHNs) obviously would have been interesting as well. However, the lack of specific driver lines covering a broad range of LHNs¹⁹ as well as the absence of “morphological landmarks” (such as the microglomerulus organization in the calyx²) to identify direct ePN bouton::LHN contacts unfortunately precludes reliable measurements of synaptic kinetics here as yet.

(Line 419-421) In the future, it will be interesting to investigate of how the distinct synaptic filtering functions executed by Unc13A and Unc13B steer the downstream signal computation processes in both, the mushroom body and importantly the lateral horn.

Minor concerns

1) Quantification in Figure S3 needs to consider the low efficiency of knockdown.

This is per se certainly a justified concern. However, we would like to point the reviewer’s attention to the fact that here we used Appl-Gal4 (to drive pan-neuronal knockdown of either isoforms) which resulted in a rather efficient downregulation (81% reduction for Unc13A and 96% reduction for Unc13B level, Supplementary Fig. 3f and 3h). We honestly do not think that our conclusions are affected by the small residual protein levels.

2) Are Unc-13s also expressed in other neurons, such as local neurons in antennal lobe, lateral horn neurons, etc? How would these neurons contribute to behavior filtering by Unc-13s?

As we have shown in previous studies which were mentioned in our manuscript^{4,9}, these proteins express in all types of neurons, with both Unc13A and Unc13B being present, though at different levels and in result Unc13A/Unc13B ratios at their active zones. How their functions contribute to behaviorally relevant synaptic filtering and computations across different neuron types is a most interesting question, which we are afraid must remain subject of future investigations, however.

3) Are Unc-13s differentially expressed in different PNs (one extreme case: 13B is expressed only in PNs that drive avoidance)?

This certainly is a per se certainly fully justified concern. We performed a very extensive anatomical study on different type of projection neurons (ePNs and iPNs; Fig. 1, Supplementary Fig. 1 and Supplementary Fig. 2). We here showed that both Unc13 isoforms express in both excitatory and inhibitory projection neurons and the colocalization study as well as STED analysis sampling from many presynaptic terminals did not provide any indication for heterogeneity between projection neurons. Nevertheless, we performed a new analysis, and measured the ratio between Unc13B and Unc13A level across different ePN boutons. As

you can see in Rebuttal Fig. 9 these data are normally distributed, means we did not observe specific subclasses concerning unc13A or unc13B expression between projection neurons.

Rebuttal Fig. 9. Unc13B to Unc13A ration across different calycal boutons are normally distributed.

References

- 1 Christiansen, F. *et al.* Presynapses in Kenyon cell dendrites in the mushroom body calyx of *Drosophila*. *J Neurosci* **31**, 9696-9707, doi:10.1523/JNEUROSCI.6542-10.2011 (2011).
- 2 Yasuyama, K., Meinertzhagen, I. A. & Schurmann, F. W. Synaptic organization of the mushroom body calyx in *Drosophila melanogaster*. *J Comp Neurol* **445**, 211-226, doi:10.1002/cne.10155 (2002).
- 3 Bohme, M. A. *et al.* Active zone scaffolds differentially accumulate Unc13 isoforms to tune Ca²⁺ channel-vesicle coupling. *Nature neuroscience*, doi:10.1038/nn.4364 (2016).
- 4 Fulterer, A. *et al.* Active Zone Scaffold Protein Ratios Tune Functional Diversity across Brain Synapses. *Cell Rep* **23**, 1259-1274, doi:10.1016/j.celrep.2018.03.126 (2018).
- 5 Voitkuhn, J. *et al.* The Unc13A isoform is important for phasic release and olfactory memory formation at mushroom body synapses. *J Neurogenet*, 1-9, doi:10.1080/01677063.2019.1710146 (2020).
- 6 Pech, U., Revelo, N. H., Seitz, K. J., Rizzoli, S. O. & Fiala, A. Optical dissection of experience-dependent pre- and postsynaptic plasticity in the *Drosophila* brain. *Cell Rep* **10**, 2083-2095, doi:10.1016/j.celrep.2015.02.065 (2015).
- 7 Abdelfattah, A. S. *et al.* Bright and photostable chemigenetic indicators for extended in vivo voltage imaging. *Science* **365**, 699-704, doi:10.1126/science.aav6416 (2019).
- 8 Bohme, M. A. *et al.* Active zone scaffolds differentially accumulate Unc13 isoforms to tune Ca(2+) channel-vesicle coupling. *Nat Neurosci* **19**, 1311-1320, doi:10.1038/nn.4364 (2016).
- 9 Voitkuhn, J. *et al.* The Unc13A isoform is important for phasic release and olfactory memory formation at mushroom body synapses. *J Neurogenet* **34**, 106-114, doi:10.1080/01677063.2019.1710146 (2020).
- 10 Brinks, D., Klein, A. J. & Cohen, A. E. Two-Photon Lifetime Imaging of Voltage Indicating Proteins as a Probe of Absolute Membrane Voltage. *Biophys J* **109**, 914-921, doi:10.1016/j.bpj.2015.07.038 (2015).
- 11 Chamberland, S. *et al.* Fast two-photon imaging of subcellular voltage dynamics in neuronal tissue with genetically encoded indicators. *Elife* **6**, doi:10.7554/eLife.25690 (2017).
- 12 Svoboda, K. & Yasuda, R. Principles of two-photon excitation microscopy and its applications to neuroscience. *Neuron* **50**, 823-839, doi:10.1016/j.neuron.2006.05.019 (2006).
- 13 Bracker, L. B. *et al.* Essential role of the mushroom body in context-dependent CO(2) avoidance in *Drosophila*. *Curr Biol* **23**, 1228-1234, doi:10.1016/j.cub.2013.05.029 (2013).
- 14 Oswald, D. *et al.* Activity of defined mushroom body output neurons underlies learned olfactory behavior in *Drosophila*. *Neuron* **86**, 417-427, doi:10.1016/j.neuron.2015.03.025 (2015).
- 15 Tsao, C. H., Chen, C. C., Lin, C. H., Yang, H. Y. & Lin, S. *Drosophila* mushroom bodies integrate hunger and satiety signals to control innate food-seeking behavior. *Elife* **7**, doi:10.7554/eLife.35264 (2018).
- 16 Wang, Y. *et al.* Blockade of neurotransmission in *Drosophila* mushroom bodies impairs odor attraction, but not repulsion. *Curr Biol* **13**, 1900-1904, doi:10.1016/j.cub.2003.10.003 (2003).
- 17 Bates, A. S. *et al.* Complete Connectomic Reconstruction of Olfactory Projection Neurons in the Fly Brain. *Curr Biol* **30**, 3183-3199 e3186, doi:10.1016/j.cub.2020.06.042 (2020).
- 18 Dolan, M. J. *et al.* Neurogenetic dissection of the *Drosophila* lateral horn reveals major outputs, diverse behavioural functions, and interactions with the mushroom body. *Elife* **8**, doi:10.7554/eLife.43079 (2019).
- 19 Frechter, S. *et al.* Functional and anatomical specificity in a higher olfactory centre. *Elife* **8**, doi:10.7554/eLife.44590 (2019).

- 20 Potter, C. J., Tasic, B., Russler, E. V., Liang, L. & Luo, L. The Q system: a repressible binary system for transgene expression, lineage tracing, and mosaic analysis. *Cell* **141**, 536-548, doi:10.1016/j.cell.2010.02.025 (2010).
- 21 Lerner, H., Rozenfeld, E., Rozenman, B., Huetteroth, W. & Parnas, M. Differential Role for a Defined Lateral Horn Neuron Subset in Naive Odor Valence in *Drosophila*. *Sci Rep* **10**, 6147, doi:10.1038/s41598-020-63169-3 (2020).
- 22 Varela, N., Gaspar, M., Dias, S. & Vasconcelos, M. L. Avoidance response to CO₂ in the lateral horn. *PLoS Biol* **17**, e2006749, doi:10.1371/journal.pbio.2006749 (2019).
- 23 Parnas, M., Lin, A. C., Huetteroth, W. & Miesenbock, G. Odor discrimination in *Drosophila*: from neural population codes to behavior. *Neuron* **79**, 932-944, doi:10.1016/j.neuron.2013.08.006 (2013).

Reviewers' Comments:

Reviewer #4:

Remarks to the Author:

The authors have addressed my concerns.